

# An automatic lake-model application using near real-time data forcing: Development of an operational forecast model for Lake Erie

Shuqi Lin[1], Leon Boegman[1], Shiliang Shan[2], Ryan Mulligan[1]

[1]Department of Civil Engineering, Queen's University, Kingston ON Canada K7L 3N6

[2]Department of Physics and Space Science, Royal Military College of Canada, Kingston ON Canada K7K 7B4.

*Correspondence to:* Shuqi Lin (shuqi.lin@queensu.ca)

**Abstract.** For enhanced public safety and water resource management, a three-dimensional operational lake hydrodynamic forecast system called COASTLINES (Canadian cOASTal and Lake forecastINg modEl System) was developed. The modelling system is built upon the Aquatic Ecosystem Model (AEM3D) model, with predictive simulation capabilities developed and tested for a large lake (i.e., Lake Erie). The open-access web-based platform derives model forcing, code execution, post-processing and visualization of the model outputs, including water level elevations and temperature, is in near real-time. COASTLINES currently generates 240-h predictions using atmospheric forcing from 15 km and 25 km horizontal-resolution operational meteorological products from the Environment Canada Global Deterministic Forecast System (GDPS). Simulated water levels were validated against observations from 6 gauge stations, with model error increasing for longer forecast times. Satellite images and lake buoys were applied to validate forecast lake surface temperature (LST) and the water column thermal stratification. The forecast LST is as accurate as hindcasts, with a root-mean-square-deviation <2°C. COASTLINES predicts storm-surge events and up-/down-welling events that are important for flood water and drinking water/fishery management, respectively. Model forecasts are available in real-time at https://coastlines.engineering.queensu.ca/. This study provides an example of the successful development of an operational forecasting system, entirely driven by open-access data, that may be easily adapted to simulate aquatic systems or to drive other computational models, as required for management and public safety.



## 1 Introduction

Lakes hold a large proportion of the global surface freshwater, which supports biodiversity and supplies water resources for drinking, transportation and recreation. However, anthropogenic stressors are causing significant changes in the properties of lakes, such as rapid warming of surface water (O'Reilly et al., 2015), major seasonal water level fluctuations (Gronewold and Rood, 2019), increased frequency of extreme events (Saber et al., 2020) and severe water quality issues such as oxygen depletion (Rowe et al., 2019; Scavia et al., 2014) and harmful algal blooms (Brookes and Carey, 2011; Watson et al., 2016). Effort has been spent on investigating the long-term responses of physical processes in lakes to climate change (O'Reilly et al., 2015; Woolway and Merchant, 2019), but improving lake monitoring and developing short-term forecast models, to predict the occurrence of extreme events is also necessary (Woolway et al., 2020). The biogeochemical cycles in lakes are complex and often regulated by physical forcing; therefore, the first step to model and forecast water quality issues, like harmful algal blooms (Paerl et al., 2011; O'Neil et al., 2012) and hypoxia (Rao et al., 2008; Rao et al., 2014), is the development of accurate hydrodynamic hindcast and forecast models.

Over the past several decades, many computational fluid dynamics models have been applied to hindcast lake hydrodynamics to aid management. These range from one-dimensional (1D) models such as DYRESM (Antenucci and Imerito, 2000), Simstrat (Gaudard et al., 2017), and GLM (Hipsey et al., 2014), to three-dimensional (3D) models such as Delft3D (Lesser et al., 2004), FVCOM (Chen et al., 2013; Gronewold et al., 2019; Rowe et al., 2019) and ELCOM (Hodges et al., 2000). Several of these hydrodynamic models are coupled to biogeochemical models to allow for prediction of water quality. In the case of hindcast applications, the complex and time-consuming setup and calibration procedure, of these models, can result in a significant time lag (months to years) between when a project is initiated and when the model results are communicated to stakeholders, which severely limits the utility of computational models for management decision making. For better application of these powerful tools, rapid monitoring and forecast systems should be established.

In addition to the significant effort required to setup and calibrate models, other hurdles exist such as data-sharing agreements between the agencies collecting forcing/validation data and those running the models. For example, the US National Oceanic and Atmospheric Administration (NOAA) Great Lakes Coastal Forecasting System (Chu et al., 2011; Anderson et al., 2018), is a comparatively large-budget multi-institutional (NOAA-GLERL and U. Michigan-CIGLR) project that predicts water levels, temperature profiles, currents, and wave heights over a 120-h timeframe in the five Laurentian Great Lakes and connecting channels, using FVCOM on a 3D unstructured grid with 30-2000 m horizontal resolution. Similarly, meteolakes.ch (Baracchini et al., 2020), applies Delft3D with short-term forecasts (4.5 days) of Swiss lakes, under a data sharing agreement between Swiss Federal Institute of Aquatic Science and Technology (EAWAG), École Polytechnique Fédérale de Lausanne (EPFL) and MeteoSwiss.

With the present online proliferation of near real-time data from lake observation buoys (e.g., https://www.ndbc.noaa.gov/; https://www.glos.us/; https://marees.gc.ca/eng/) and high-resolution meteorological forecasts (https://dd.weather.gc.ca/model_gem_global/), data collection, assembly of forcing files, model execution, post processing and online communication of model results can be automated to near real-time, without a need for data-sharing arrangements. This drastic improvement in workflow efficiency can allow for the development of





specific simulations tailored to meet diverse lake-management needs (e.g., high-resolution nearshore grids, spill
modelling, fisheries research, beach closures, and optimization of treatment or source water monitoring strategies).
In the present study we develop a pilot operational lake forecasting system for Lake Erie. The system is called
COASTLINES (Canadian cOASTal and Lake forecastINg modEl System) and it uses Python-based wrapper code,
that processes publicly available real-time data to execute a hydrodynamic lake model and produce web-based real-
time products to communicate the results. The objective of this paper is to assess the accuracy of the model in
forecasting water levels and temperature fields, compared to traditional hindcast applications of numerical models.
This will determine the reliability of the model for short-term water management decision support for government
agencies and other stakeholders. The data and methods are presented in Section 2, with an overview of
COASTLINES including the workflow and a description of the implementation for a large lake (i.e., Lake Erie). The
results are described in Section 3, including validation and evaluation of the forecast variables (water levels and
temperatures), showcasing the short-term predictive ability of COASTLINES over timescales of 24-h and 240-h. A
discussion of the forecast performance of COASTLINES with other operational platforms of lakes (GLCFS,
meteolakes.ch) and hindcast simulations is provided in Section 4, including an analysis of the advantages and
potential bias and uncertainty.

## 2    Data and methods

### 2.1    Study site

Lake Erie, the shallowest lake of the Great Lakes with a mean depth of 19 m. Lake-wide hydrodynamics
predominantly exhibits free surface and current oscillations from the 14-h barotropic seiche (Hamblin 1987;
Boegman et al., 2001).  The lake morphometry consists of distinct, yet interconnected western, central, and eastern
basins (Fig. 1), each with its own water quality concerns: The 11-m deep western basin is typically well mixed and
has frequent harmful algae blooms related to climate-driven meteorological forcing (Michalak et al., 2013).  The
ephemeral stratification in late summer (Loewen et al., 2007) regulates vertical biogeochemical fluxes (Boegman et
al., 2008). The 20-m deep central basin is prone to large-scale hypolimnetic hypoxia (Scavia et al., 2014).
Hydrodynamics are governed by an internal Poincaré wave (Bouffard et al., 2012; Valipour et al., 2015) and a bowl-
shaped depression of the summer thermocline, which influence the oxygen budget (Beletsky et al., 2012; Bouffard et
al., 2014).  The 65-m deep eastern basin has nearshore water quality concerns from *cladophora* (Higgins et al.,
2006) and ecosystem engineering by dreissenid mussels (Hecky et al., 2004).  Hydrodynamics of this region are
controlled by the coastal internal Kelvin wave (Valipour et al., 2019).



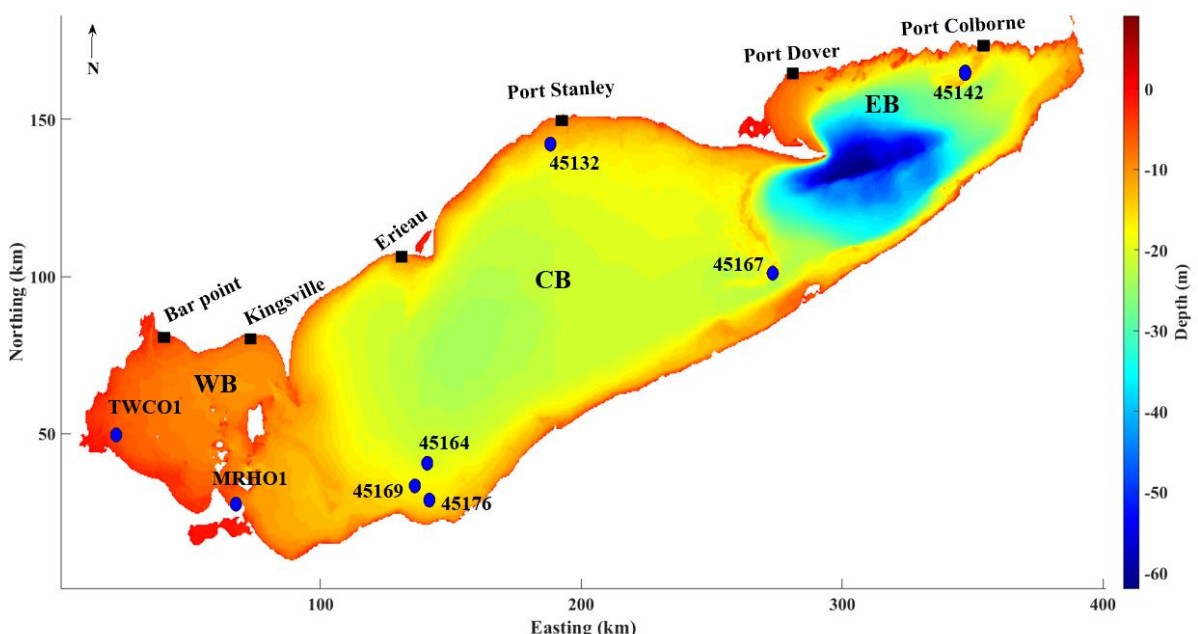

**Fig.1 Map of Lake Erie showing the bathymetric depths and observation sites. The bathymetric map is at the**
**resolution of the 500 m grid applied in the model. The western, central, and eastern basins are labeled as WB,**
**CB, and EB, respectively. Blue circles indicate lake buoys and black squares indicate water level gauges.**
**2.2      Model description**
COASTLINES applies the three-dimensional Aquatic Ecosystem Model (AEM3D, HydroNumerics Pty Ltd.). This
model solves unsteady 3D Reynolds-averaged Navier-Stokes equations for incompressible flow based on
Boussinesq and hydrostatic approximations. The advection of momentum in the model is based on the Euler-
Lagrange method with a conjugate-gradient solution for the free-surface height (Casulli and Cheng, 1992), and a
conservative ULTIMATE QUICKEST discretization scheme is used for advection of scalars (Leonard, 1991).
AEM3D is a parallel version of the commonly applied Estuary and Lake Computer Model (ELCOM; Hodges et al.,
2000). ELCOM has been applied to Lake Erie to simulate currents and seasonal circulation (León et al., 2005), the
internal Poincaré (Valipour et al., 2015) and Kelvin waves (Valipour et al., 2019), ice cover (Oveisy et al., 2012)
and the response of the thermal structure, in Lake Erie, to air temperature and wind speed changes (Liu et al., 2014).
ELCOM has been coupled with the biogeochemical CAEDYM model to simulate Lake Erie phytoplankton and
nutrients (León et al., 2011), the response of hypoxia (Bocaniov and Scavia 2016) and algae blooms (Scavia et al.,
2016) to nutrient load reductions.  Recent applications of AEM3D include a study of the water level in Lake
Arrowhead, California (Saber et al., 2020), ice cover in Lake Constance (Caramatti et al., 2019) and pollutant
transport in Lake St. Clair (Madani et al., 2020).



### 2.3 Model setup and meteorological forcing variables

To adequately resolve the coastal boundary layer (~ 3 km width; Rao and Murthy, 2001) and basin-scale internal waves (Poincaré (16.8 h) and Kelvin waves), the bathymetry of Lake Erie (https://www.ngdc.noaa.gov/mgg/greatlakes/erie.html) was discretized into a 500 m × 500 m horizontal grid, which is ~10 % of the internal Rossby radius (Schwab and Beletsky, 1998). The lake was discretized into 45 vertical layers, with fine resolution (0.5 m) through the surface layer, metalimnion and bottom of the central basin, and coarse layers (5 m) through the hypolimnion of the deeper eastern basin to the maximum depth of 64 m. The model was 'cold started' with the surface water temperature observed at station 45142 and MHRO1 on day of year (day) 99, 2020, at a time when the spring turnover and stratification is minimal, and the model has been running continuously since that time. The model time step is dt = 300 s to satisfy the CFL (Courant-Friedrichs-Lewy) condition for internal waves, which is CFL = (Hodges et al., 2000).

The model is driven by meteorological forcing including wind speed, wind direction, air temperature, shortwave solar radiation, relative humidity, air pressure, and net longwave radiation. The net longwave radiation is computed internally within AEM3D from cloud cover and modelled surface temperature. In order to address the spatial variability of meteorological conditions across the lake, the computational domain was forced with meteorological data on horizontal grids at 15 km (https://dd.weather.gc.ca/model_gem_global/15km/ ) and 25 km (https://dd.weather.gc.ca/model_gem_global/25km/) resolution using meteorological forecasts from the Environment and Climate Change Canada Global Deterministic Forecast System (GDPS). This resulted in 31 and 23 meteorological sections for the 15 km and 25 km models, respectively. Wind speed, direction, air temperature, relative humidity, air pressure, dew point, and cloud cover are direct outputs from GDPS, with solar radiation calculated based on dew point and air pressure (Meyers and Dale 1983; Appendix C. in Gaudard et al., 2019). The meteorological forecast has an output timestep of 3-h and a forecast length of 240 hours. The .GRIB2 meteorological data were retrieved via 'urllib' library in Python and formatted into AEM3D input files using the nctoolbox in MATLAB.

In this pilot application, the Lake Erie inflows and outflows, which roughly balance, are neglected, however evaporation and precipitation are accounted for in the water balance.

### 2.4 Observations, implementation and model validation

The water levels and temperatures simulated by COASTLINES were validated using both in situ and satellite observations. Near real-time water level data was used from six stations along the Canadian coastline, which reported hourly observations (Bar Point, Kingsville, Erieau, Port Stanley, Port Dover, and Port Colborne; Fig. 1; Table 1), retrieved from Fisheries and Oceans Canada (https://marees.gc.ca/eng/find/zone/44). The data are parsed using the 'BeautifulSoup' library in Python and saved as .csv files (Appendix A1), to be read with MATLAB for model validation. The observations showed higher fluctuations in the western (Bar Point and Kingsville) and eastern (Port Dover and Port Colborne) basins (Fig. 1). Thus, we quantify the water level forecast capability in terms of the Root Mean Square Deviation (RMSD) and Relative Error (RE):





$RMSD = [\frac{1}{N}\sum_{i=1}^{N}(x_i - y_i)^2]^{1/2},$  (1)
$RE = 100\frac{RMSD}{log.\ mean(daily\ range)},$  (2)
where $x_i$ and $y_i$ ( $i$ = 1, 2, 3, ... $N$) are the model and observed water level timeseries and $N$ is the number of samples.
RMSD is the absolute error of the model against the observation. The difference between the observed daily
minimum and maximum value was defined as the daily water level fluctuation range, and the RE is the ratio
between RMSD and lognormal mean of daily range over April to September 2020. Given that our study focusses on
a 240-h forecast, RE is able to characterize the forecast bias, regardless of the instantaneous water level position.
Eight in situ lake buoys, distributed over the nearshore areas of three basins (Fig. 1; Table 1), provided near real-
time data through the Great Lakes Observing System (GLOS: https://www.glos.us/) and National Data Buoy Center
(NDBC: https://www.ndbc.noaa.gov/) portals. The text-based NDBC observations in are parsed using the
'BeautifulSoup' Python library (Appendix A2), and the GLOS observations are retrieved using 'webdriver' from the
'selenium' Python library. All the lake buoy observations are saved as .csv files and read into MATLAB for post-
processing. This process is repeated for each station. Attempts to retrieve missing variables results in run-time
errors.
The lake buoys are deployed from April or May through mid-October, spanning the spring/fall turnovers and
seasonal summer stratification periods. However, due to COVID-19 related delays in instrument deployments in
2020, only two buoys located offshore of Cleveland near the water intake crib (station 45176 and station 45164)
were equipped with thermistor chains to monitoring temperature profiles.  The other six buoys provide air and
surface water temperature as well as wind speed and direction observations applied for lake surface temperature
(LST) and meteorological forecast validation. Satellite-based observations of LST were obtained from the Great
Lakes Surface Environmental Analysis (GLSEA2), which is derived from NOAA CoastWatch AVHRR (Advanced
Very High-Resolution Radiometer) imagery and updated on NOAA GLERL website
(https://coastwatch.glerl.noaa.gov/erddap/files/GLSEA_GCS/). GLSEA2 produced daily observations, with 2.6 km
resolution, from the cloud-free portions of the satellite images (Schwab et al., 1999). The data are in netCDF format,
which is retrieved using the 'BeautifulSoup' library and 'webdriver' from 'selenium' (Appendix A3).
We quantify the temperature forecast capability using the statistical measures of RMSD (eq. 1) and Mean Bias
Deviation (MBD):
$MBD = 100\frac{\frac{1}{N}\sum_{i=1}^{N}(x_i - y_i)}{\frac{1}{N}\sum_{i=1}^{N}y_i}$  (3)
In *spatial* MBD and RMSD (s-MBD and s-RMSD), $x_i$ and $y_i$ are the model and observed temperature in each grid,
and $N$ is the total number of grids. In *timeseries* MBD and RMSD (t-MBD and t-RMSD), $x_i$ and $y_i$ are the model and
observed temperature at each sample time, and $N$ is the total number of samples.
**Table 1**
**Details of field stations with water level gauges and lake buoys.**

| Station | Parameter | Sampling interval (min) | Depth of measurement (m) |
|---------|-----------|-------------------------|--------------------------|
| Bar Point | Water level | 60 | Surface |
| Kingsville | Water level | 60 | Surface |





| Erieau | Water level | 60 | Surface |
|---|---|---|---|
| Port Stanley | Water level | 60 | Surface |
| Port Dover | Water level | 60 | Surface |
| Port Colborne | Water level | 60 | Surface |
| TWCO1 | Temperature | 10 | Surface |
| 45005 | Temperature | 10 | Surface |
| 45176 | Temperature | 10 | 1, 3, 4, 6, 7, 9, 10, 12, 14, 15 |
| 45169 | Temperature | 30 | surface |
| 45164 | Temperature | 60 | 1, 2, 4, 6, 8 10 |
| 45132 | Temperature | 60 | Surface |
| 45167 | Temperature | 10 | Surface |
| 45142 | Temperature | 60 | Surface |


## 2.5 System operation

The COASTLINES operational forecast system is run on a local server supported by Queen's University (Kingston,
Canada). The COASTLINES workflow is presented in Fig. 2. The system consists of input data acquisition and
preparation, 24-h hydrodynamic simulations, 240-h hydrodynamic simulations, validation against in situ
observations, and uploading the model forecasts and validation to the web platform. Given that the standard
deviations of meteorological forecast variables increase with forecast lead time (Buehner et al., 2015), we separated
the 24-h and 240-h forecast simulations, with both performed daily. The model advances daily according to the 24-h
forecast simulation and generates 're-start' files. These files are then used to initiate 240-h forecast simulations and
the 24-h simulations for the next day. The input files for the new 240-h forecast simulations are replaced by the new
input files with the 240-h meteorological forecast generated each day. Daily 24-h and 240-h forecast model outputs
are compared against observations, respectively, to evaluate the forecast performance against forecast length.
Automation of the processing tasks in the system is performed by Python scripts triggered by the Windows Task
Scheduler every 24-h at midnight. The online meteorological forecast data are retrieved from GDPS once updated at
around 3 am EST. Forcing variables are formatted in MATLAB, called by the Python scripts once the
meteorological forecast data from GDPS are retrieved. The 24-h simulation and 240-h simulations take 0.5 h and 4 h
to complete, respectively, on a 32-core XEON workstation. The observed data, including water level from gauge
stations, water temperature from lake buoys and satellite imageries are scraped at 8 am, followed by post-processing
in MATLAB to validate model output, calculating statistical metrics (RMSD, MBD). The results are exported to
Google sheets and published to the COASTLINES website (e.g., Appendix B). Global coverage of the GDPS
forecasts enables this operational system to be readily implemented at other sites where lake bathymetry, boundary
flows and in-situ validation data are available.



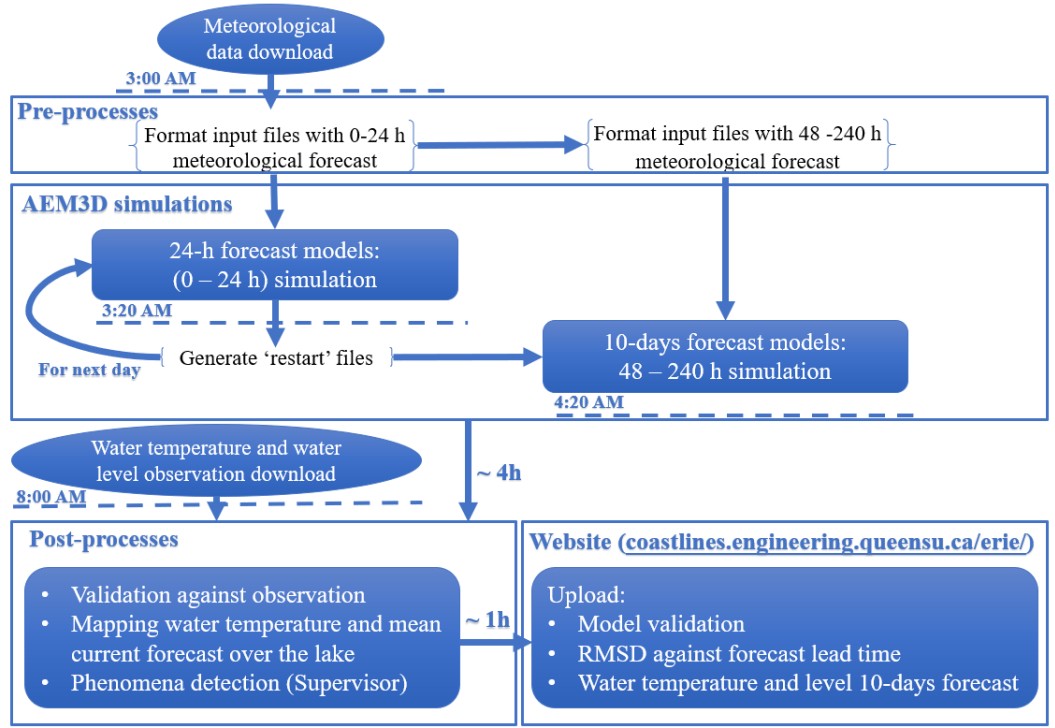


**Fig. 2 Daily Python workflow and automated processes in the COASTLINES operational system as performed**
**on the local server.**


**3      Results**
The COASTLINES water level and temperature forecasts have been operational since April and July 2020,
respectively. The 24-h and the 240-h forecast of water levels from the 15 km and 25 km resolution models were
validated against real-time gauge station observations. The statistical metrics of water level RMSD and RE were
ensembled over April to September 2020. The 24-h and the 240-h forecast of LST and temperature profiles from the
models were also validated against real-time lake buoys and daily averaged satellite imageries. The timeseries and
spatial MBD and RMSD (t-RMSD, t-MBD and s-RMSD, s-MBD) were ensembled over July to September 2020.
**3.1      Water level**
The Relative Error (RE) of the forecast water level generally increases with forecast time when averaged over April
to September 2020; the 24-h forecast error being ~ 40% at all six gauge stations (Fig. 3 a, c, e, g, i, k). Given the
large water level fluctuation at Port Colborne (Fig. 3 l), the 240-h forecast RE is highest at this station, exceeding
70% (Fig. 3 k). Of the six gauge stations reported in this study, those at the western (Bar Point and Kingsville) and
eastern (Port Dover and Port Colborne) ends of Lake Erie longitudinal axis had the largest water level fluctuations,
resulting from the predominant south-westerly winds generating strong wind set-up and surface seiches (Fig. 3 b, d,



f, h, j, l). The lognormal means of the daily range in water level at the six gauge stations are 0.21 cm (Bar Point),
0.16 cm (Kingsville), 0.07 cm (Erieau), 0.10 cm (Port Stanley), 0.15 cm (Port Dover), 0.17 cm (Port Colborne).
The 24-h forecasts show qualitative agreement with observations in phase and magnitude (Fig. 4). The 24-h
forecasts reproduce the dramatic surface seiches induced by westerly winds > 15 m s$^{-1}$ (Fig. C2) on day 251 (RMSD
< 0.1 cm), especially the obvious water level fluctuations at stations in the western and eastern basins (Fig. 4 a, b, e).
However, the prediction of water level at Bar Point showed large bias (Fig. 4 f), with the model overestimating the
decrease in water level. This error may result from neglecting the large Detroit River inflow, which occurs near Bar
Point. The uncertainty in the model forecast, which increased with the range of the daily fluctuation, was captured
by the ensemble 24-h forecast RE over April to September (the shaded areas in Fig. 4). Overall, the confidence
interval of the 24-h forecast can include most of the discrepancies between the observations and the model results.

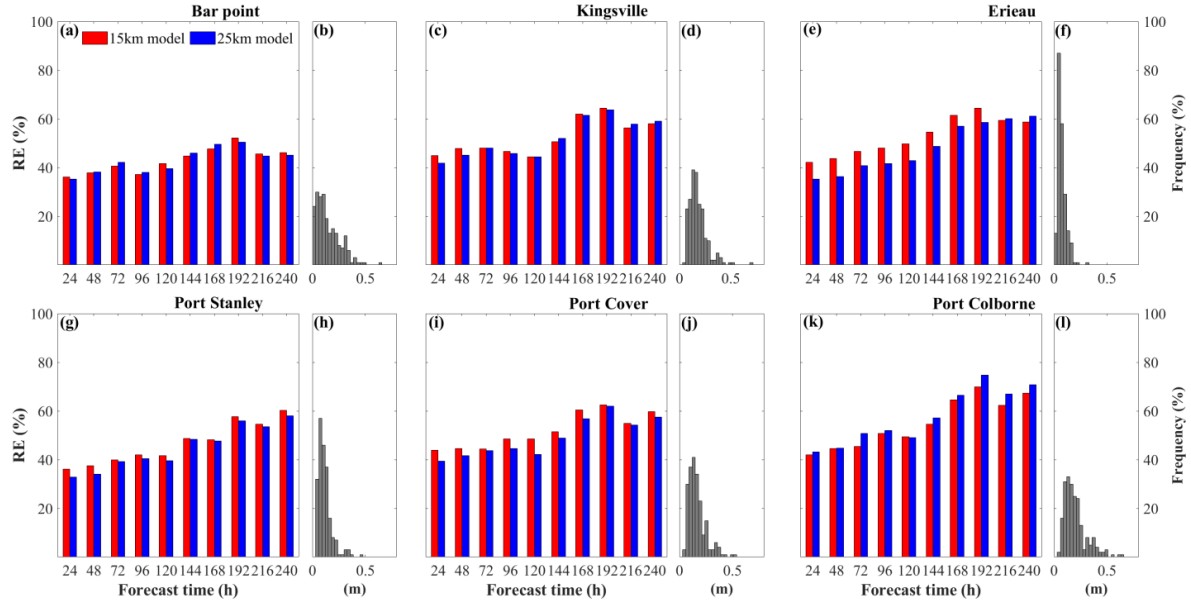

**Fig. 3 Relative error (RE) in water level predictions against forecast time at six stations (a, c, e, g, i, k). Panels**
**(b, d, f, h, j, l) are the corresponding frequency distribution of lognormal means of the daily water level**
**fluctuation range (x-axes, unit in meter) at Bar Point, Kingsville, Erieau, Stanley, Port Dover, Port Colborne,**
**respectively.**

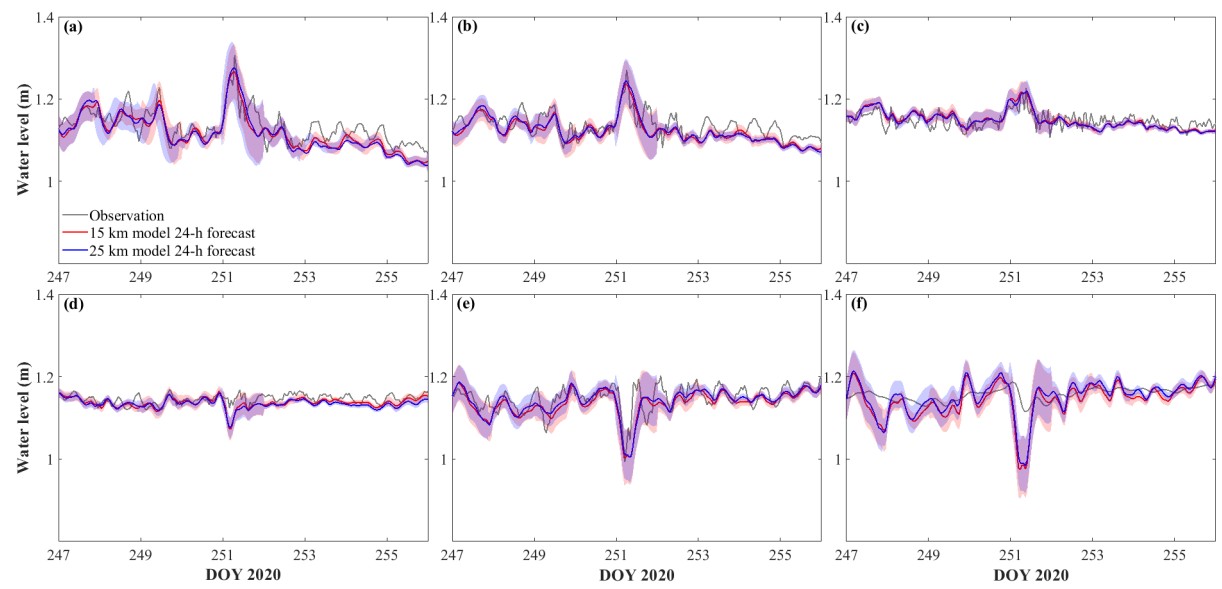

**Fig. 4 Comparison between observed and stitched 24-h forecast modeled water level at (a) Port Colborne, (b) Port Dover, (c) Port Stanley, (d) Erieau, (e) Kingsville, and (f) Bar Point. The shaded areas show the confidence interval of the 15 km model (red shading) and the 25 km model (blue shading), as given by the ensemble 24-h RE in Fig. 3.**

Timeseries validations for the 240-h model forecast (Fig. 5) include confidence intervals from the ensembled RE (Fig. 3). As shown, the forecast began 6 days in advance of the large surface seiche event on day 251 and predicted the seiche to crest at Port Colborne 1-2 h ahead of the observations, and to trough at Kingsville 1-2 h behind the observations (Fig. 5 a, c). Damping of the seiche oscillations (~144 hours in the future) was excessive, with the water levels being underestimated and the phase shifted by approximately 12 hours (Fig 5. a, b). Despite the wide confidence intervals, due to the increasing RE with forecast time, large bias existed after the seiche event (forecast time >168 hours). When the forecast initiation was close to the event (3 days before), the prediction of seiche phase was more accurate (Fig. 5 d, e, f). However, the seiche decay still had a 12-h phase shift. The discrepancies in seiche amplitude (< 0.1 m) were within the confidence intervals of the models.

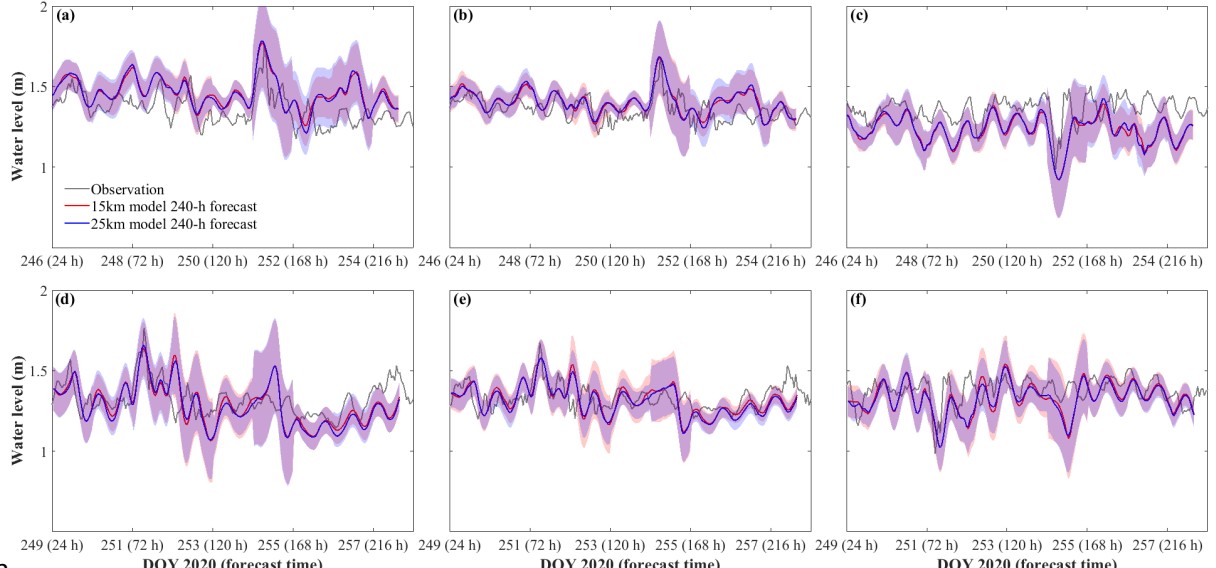

**Fig. 5 Comparison between the observed water level and 240-h forecast initiated on day 245 (a, b, c) and day 248 (d, e, f) at Port Colborne, Port Dover, and Kingsville, respectively. The shaded areas show the confidence interval of the 15 km model (red shading) and the 25 km model (blue shading), as given by the ensemble 240-h RE in Fig. 4.**

### 3.2 Water temperature

#### 3.2.1 Lake Surface temperature

Using satellite-based and lake buoy-based observations, we evaluated the LST forecast (Fig. 6). The 24-forecast captured the seasonal variation of LST, particularly the rapid increase in temperature on days 180-190, and the gradual decrease in temperature after day 240; at all eight stations. However, the forecast overestimates the LST in July with 3-5 °C (days 180-210), especially at STN 45167 and 45142. Due to the 3-h delivery interval associated with the meteorological forecast data, the forecast model was insensitive to temperature fluctuations over shorter timescales, as recorded by the lake buoys, and it underestimated the sudden decrease in temperature near day 220 and 255 at STN 45005.

Overall, the t-MBD and t-RMSD, over these eight stations, were ~6% and 1.4 °C (15 km model) and ~5% 1.3 °C (25 km model), respectively (Table 2). The average s-MBD and s-RMSD over the 50 days from July-September were ~4% and 1.2 °C, respectively, for both 15 km and 25 km resolution models.



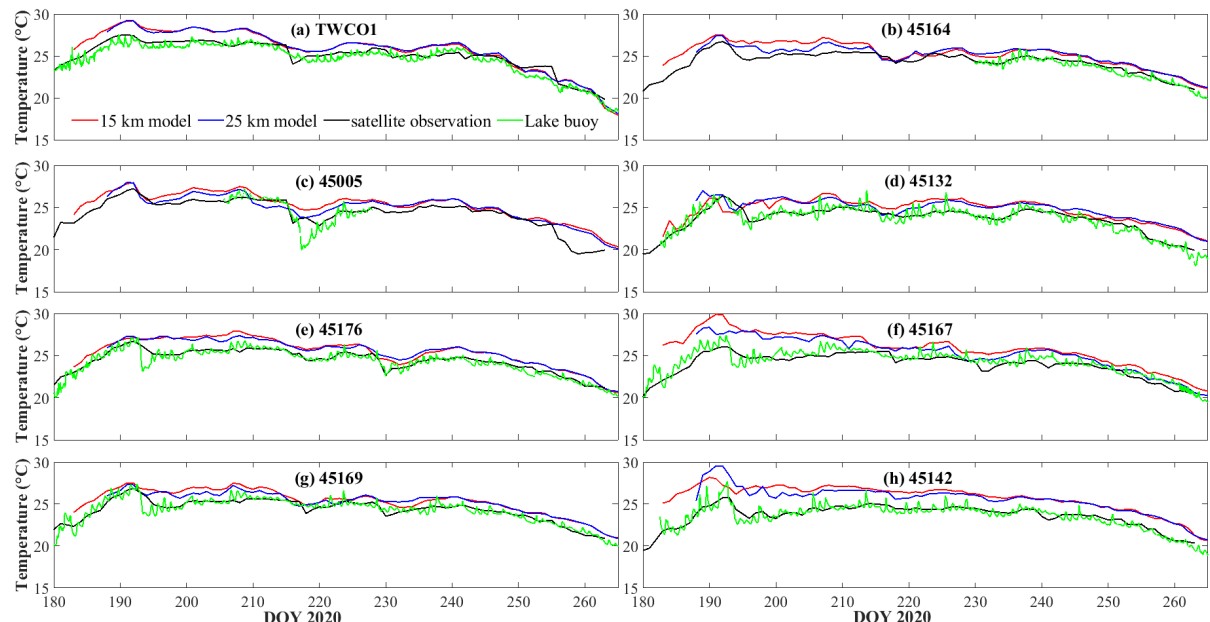

**Fig. 6 Comparison between the stitched 24-h forecast and observed lake surface temperature (LST) at 8 stations**
**(a) TWCO1, (b) 45164, (c) 45005, (d) 45132, (e) 45176, (f) 45167, (g) 45169, and (h) 45142. The green lines are**
**timeseries observations from lake buoys, the black lines are daily observations derived from satellite imagery.**

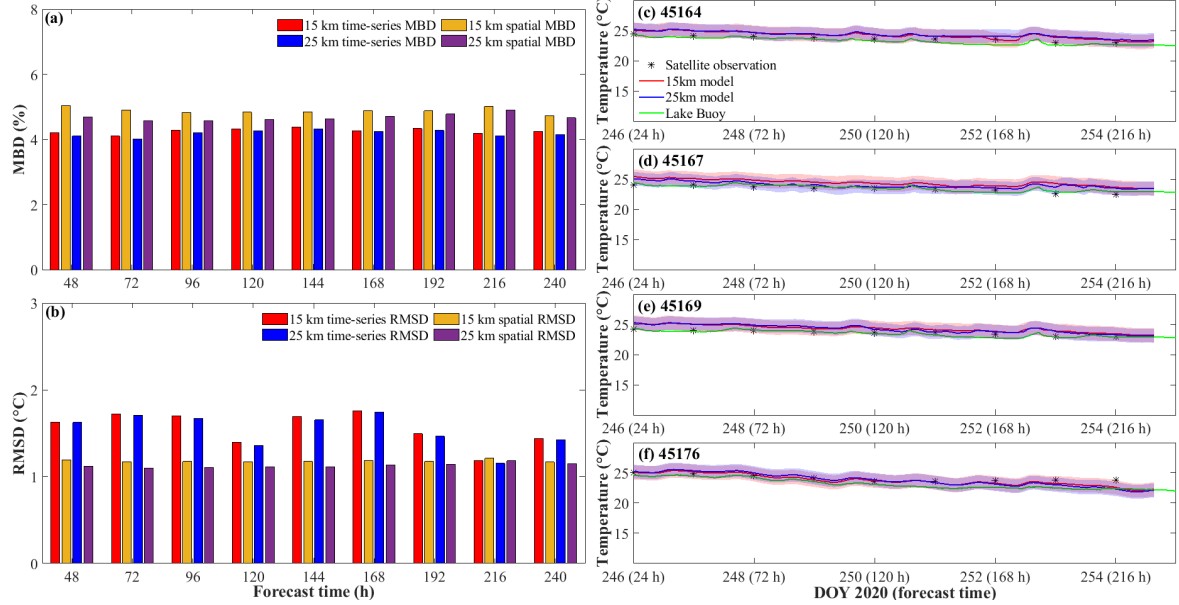

**Fig. 7 (a) Mean-Bias Deviation (MBD) against forecast time; (b) Root-Mean-Square Deviation (RMSD) against**
**forecast time. (c-f) Timeseries of 240-h forecast and observed LST at stations 45164, 45167, 45169, 45176,**
**respectively, and daily averaged satellite LST (black asterisks). The confidence interval (shaded areas) in (c-f)**



**represents the uncertainty of the 240-h forecast model through the timeseries RMSD with the forecast time**
**(panel b).**
The 240-h forecast MBD and RMSD, surprisingly, do not show an increase in error with forecast time (Fig. 7 a, b).
Both t-MBD and s-MBD, over the 240-h forecast, are ~4-5%, with s-MBD 0.5-1% higher than t-MBD. Although
both 240-h s- and t-RMSD are under 2 ℃, the t-RMSD show the fluctuation with forecast time to be higher than s-
RMSD. Both timeseries observations from lake buoys and daily averaged observations from satellite imagery fall
into the forecast confidence interval based on the 240-h t-RMSD (Fig 7 c-f).
Spatial comparisons of satellite-based observations and to the 24-h, 48-h, 120-h, 168-h surface temperature forecasts
illustrate that the forecast system captured the cooling of the lake surface in late summer (Fig. 8). Without river
inputs, which adjust more rapidly to air temperatures (~3 d) compared to deeper lake waters, the model predicted
lower surface temperatures in coastal regions of the western basin, compared with the satellite observations (Fig. 8
e, f, g, i, j, k). The 24-h and 48-h forecast showed cold water along the northwest shoreline of the central basin with
a cold bias ~ 2 ℃; this may be up-welling hypolimnetic water (see following Discussion 4.2). Further comparisons
between model predictions and satellite-based observations of LST can be found in the Supporting material (Fig.
D1-2).

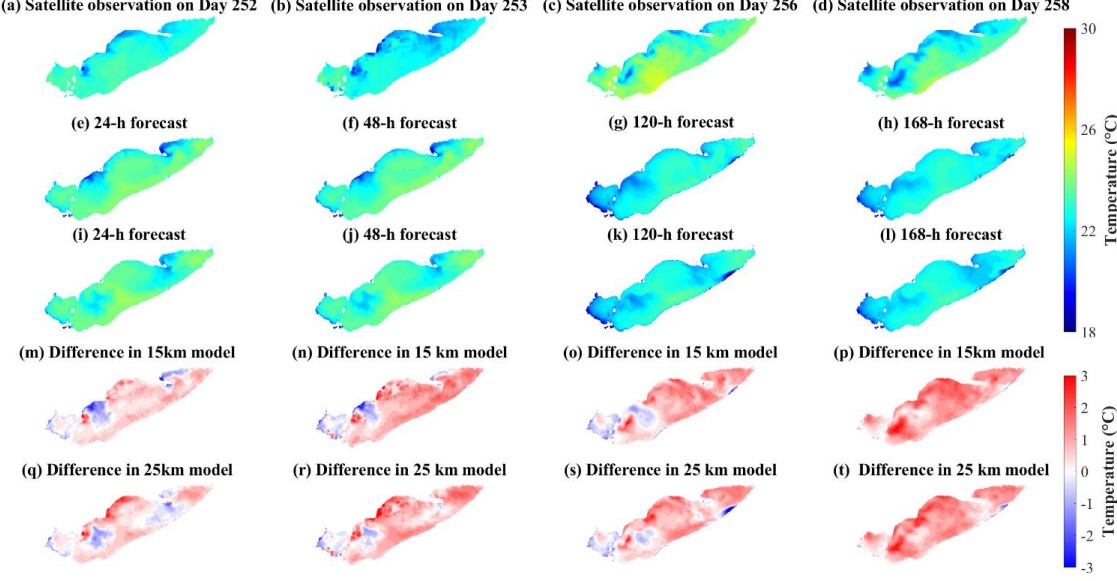

**Fig. 8 Comparison of lake surface temperature from (a-d) satellite observations, (e-h) 15 km model forecast,**
**and (i-l) 25 km model forecast during late summer. The models were initiated on day 251 The difference**
**between observations and models are shown in (m-t).**
### 3.2.2    Thermal structure
The three-dimensional structure of the AEM3D model applied in COASTLINES enables the prediction of the thermal
structure in the lake. On 15 Jun. 2020 (day 168), a rapid drop (~ 6℃) in surface temperature, was recorded by the





thermistor at STN 45176, and predicted by the stitched 24-h COASTLINES model (15 km grid) (Fig. 9 a, b). The
timing and intensity of this up-welling event were accurately forecast, but before and after the upwelling event, the
mixed layer depth was modelled to be deeper than observed; perhaps a result of spurious numerical diffusion resulting
from the thermocline swashing along the stair-step z-level grid at the lake perimeter. The 240-h forecast model was
not yet operational at this time.
Both the 240-h 15 km and 25 km resolution forecasts predicted the down-welling event on 11 Jul. 2020 (day 193) at
STN 45176 (Fig. 10). The forecasts were initiated 7 days before the event (day 187), successfully predicting when
warm surface water down-welled toward the bed, displacing the thermocline (Fig. 10 b, c), but the 15 km resolution
underestimated the intensity of this down-welling, predicting thermocline recovery on day 193. The forecast initiated
5 days before the event (day 189) presented a more accurate prediction with the down-welling persisting over 2 days
(Fig. 10 d, e) – as observed (Fig. 10 a).

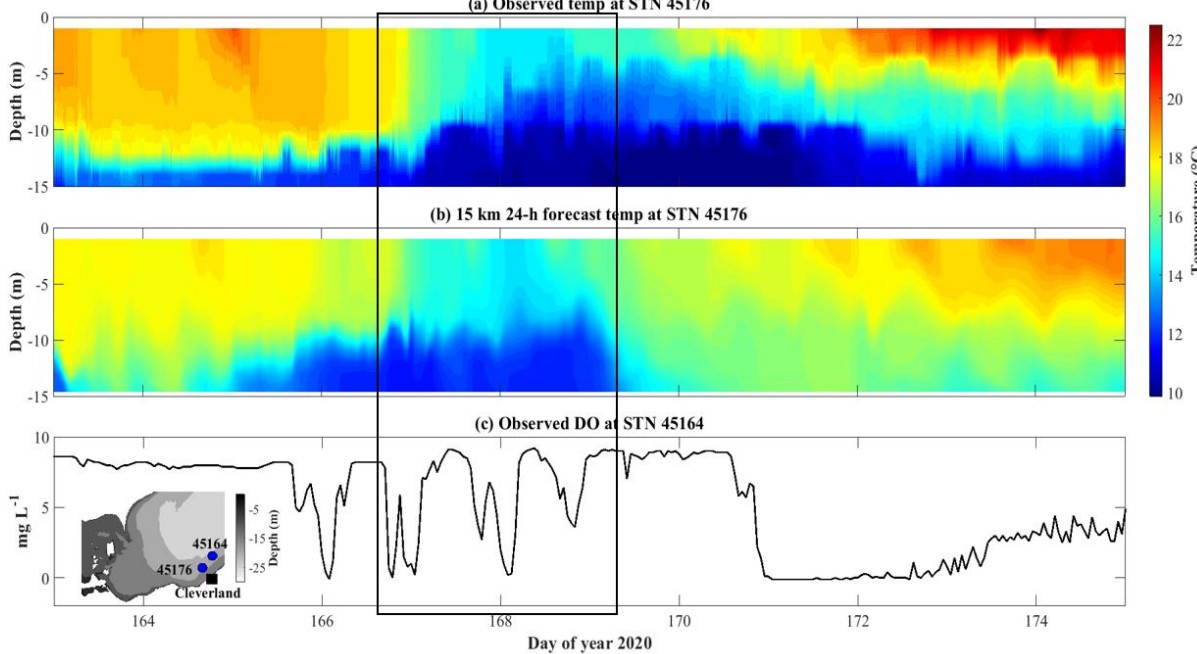

**Fig. 9 Temperature profile comparisons between (a) observations and (b) stitched daily 24-h forecasts from the 15 km resolution model at station 45176. (c) Observed dissolved oxygen concentration at station 45164 from lake buoy (https://www.glos.us/). The inset image shows the bathymetry and locations of lake buoys. The black square indicates the timing of the up-welling event.**



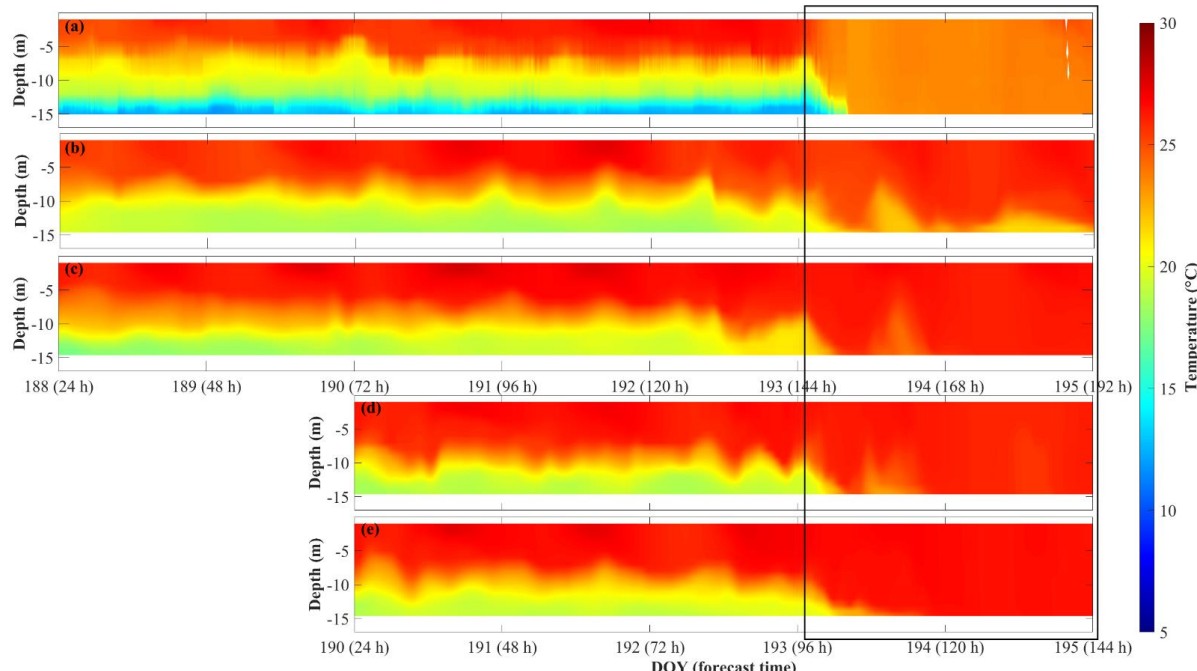

3--

**Fig. 10 Comparisons of (a) observed temperature profile, (b, d) 240-h 15 km resolution modeled, and (c, e) 240-h 25 km resolution modeled temperature profiles at STN 45176.  The forecast models were initiated on day 187 (b, c), and day 189 (d, e). The black square indicates the down-welling event.**

**Table 2**
**Statistical measures of t-MBD (Mean-Bias Deviation) and t-RMSD (Root-Mean-Square Deviation) between the 24-h forecast model and observations of water temperature.**

| Station | RMSD (°C) | | MBD (%) | |
|---|---|---|---|---|
| | 15 km model | 25 km model | 15 km model | 25 km model |
| 45176 | 2.6 | 2.6 | 6.8 | 6.8 |
| 45164 | 1.8 | 2.1 | 2.2 | 2.3 |
| 45132 | 1.5 | 1.5 | 5.5 | 5.7 |
| 45142 | 2.4 | 2.1 | 9.9 | 8.8 |
| 45167 | 1.2 | 1.1 | 4.6 | 4.0 |
| 45169 | 1.3 | 1.2 | 4.7 | 4.6 |
| TWCO1 | 1.0 | 1.0 | 3 | 1.9 |
| 45005 | 1.2 | 1.1 | 8.2 | 7.9 |

## 4 Discussion

### 4.1 Bias and uncertainty

The 240-h COASTLINES forecast is longer than the other operational lake forecast systems (GLCFS and meteolakes.ch) and is the only one forced with open-access meteorological data that has global coverage. GLCFS provides 48-h water level forecasts with RMSD ~0.12 m at the Buffalo gauge and ~0.14 m at the Toledo gauge, corresponding to RE ~ 60% and 51%, respectively (O' Connor et al., 1999; Trebitz, 2006); using the older 4 km grid





implementation of POM, as opposed to the newer unstructured grid FVCOM GLCFS. COASTLINES gives better
48-h forecast performance (RE ~ 40 %) for water levels at six gauge stations.
Benefitting from a smaller domain, finer resolution meteorological input (~2.2 km) and data assimilation, the 4.5-
day LST predicted by meteolakes.ch has RMSD = 0.8 °C (Baracchini et al., 2020), whereas COASTLINES predicts
the 120-h (5 d) LST with RMSD ~ 1.7 °C. Given this small improvement in LST prediction, it is not clear if the
added model complexity and computational cost, associated with data assimilation, justify a small improvement in
simulated water temperature; particularly, when the objective of the present work is to develop a simple automated
lake modelling system that can be readily to diverse field sites to suit management needs.
The AEM3D (formerly ELCOM) model employed in COASTLINES has shown skill in temperature hindcasts in the
Great Lakes with RMSD ~ 0.9 – 3 °C in Lake Erie (Liu et al., 2014; Oveisy et al., 2012) and 1.5 – 1.9 °C in Lake
Ontario (Paturi et al., 2012). The 24-h COASTLINES forecast predicts the water temperature with an average s-
RMSD and t-RMSD < 2 °C at the surface (Table 2). Therefore, the forecasts are within ~1 °C RMSD in comparison
to hindcasts, showing sufficient model skill for predictive simulations to aid lake management (e.g., movements of
hypoxic water, fish thermal habitat, etc.).
The accuracy of the COASTLINES forecasts, relative to hindcasts using observed meteorological conditions, may
result from the limited spatial resolution associated with historical meteorological data. Liu et al., (2014) applied
uniform Lake Erie meteorological forcing over 4 zones and Valipour et al., (2019) utilized 6 zones, each spanning
~100 km. These included land-based observations, when there was no available lake buoy data, which induces error,
especially in large shallow lakes (Hamblin, 1987). The comparatively high-resolution GDPS meteorological forecast
was four to five times higher in horizontal resolution than used in the hindcast simulations, improving the
representation of regional meteorological and climatological conditions in the model. For example, a spatially
variable wind field is essential for simulating the mean surface circulation (Laval et al. 2003). In Lake Erie, the
thermocline depth and hypolimnetic water temperature are sensitive to wind (Beletsky et al. 2012; Liu et al., 2014).
The 3-h time interval between GDPS forecast dataset updates is much less than the 10-min interval associated with
meteorological collected by lake-buoys for hindcasts (e.g., Leon et al., 2005) and so the coarse GDPS forecast
resolution may alias temporal events, such as wind gusts (Fig. C1), inducing a potential source of bias and
uncertainty in the hydrodynamic predictions. This is of particular concern in large shallow lakes, such as Lake Erie,
where winds play the dominant role in driving water level fluctuations.
Comparisons between observed and forecast meteorological data at selected stations are shown in Appendix C (Fig.
C1-5). The 24-h air temperature and wind speed forecasts had ~ 1.5 °C and ~ 2 m s$^{-1}$ RMSD, respectively. However,
in the 240-h forecast, the bias in meteorological forecast data, especially the wind forecast, increases with forecast
time (Buehner et al., 2015). The 168-h forecast meteorological data overestimates wind speeds by up to 10 m s$^{-1}$
(Fig. C4).
In addition to inaccuracy in meteorological forecasts, the discrepancies in simulating temperature profiles forecast
may result from numerical diffusion arising due to the discrete nature of the vertical and horizontal grids. The
simulated thermocline depth is overestimated (Fig. 9, 10), as occurred in applications of ELCOM with both higher
(Nakhaei et al., 2019) and lower resolution (Paturi et al., 2012). COASTLINES has the potential to predictively



simulate mesoscale physical processes, such as Kelvin waves (Bouffard and Lemmin, 2013; Valipour et al., 2019)
and nearshore-offshore exchange (Valipour et al., 2019); however model performance is poor in nearshore areas,
where topographic features remain poorly resolved (; their figure 3.14).

**4.2    Prediction of coastal up-welling for fishery and drinking water management**

The central basin of Lake Erie is vulnerable to hypoxia in the summer due to the thermal stratification and relatively
large ratio of surface area to hypolimnetic volume. Associated fish kills events (10s of thousands) are regularly
reported, including an event on north shore of the central basin in the late summer of 2012, which was presumably
was caused by up-welling of cold anoxic water from the hypolimnion (MOE, 2012; Rao et al., 2014). Similarly,
1000s of freshwater drum were killed in a rapid warming event (~5 °C /week) in the western basin in 2020
(https://www.13abc.com/content/news/Hundreds-of-dead-fish-wash-up-in-Sandusky-Bay-571025541.html).
Shoreward advection of hypoxic water, from up-welling or internal waves also adversely affects source water
quality at drinking water intakes (https://epa.ohio.gov), whereby high Fe and Mn or low pH, associated with hypoxia
water require adjustments to treatment processes.  This is particularly an issue along the Ohio coast of the central
basin (Ruberg et al., 2008; Rowe et al., 2019).
The ability to predict these movements of hypolimnion water would aid management of both the Lake Erie fisheries
and drinking water treatment.  Here, we test the ability of the model to predict up-welling of cold bottom water in
the region where the fish kill was observed in 2012. On days 249-253, 2020 (Fig. 8) strong southwesterly winds (~
12 m s$^{-1}$; Fig. C2) were modelled and observed to create up-welling along the north shore, as expected from Ekman
drift of the surface layer. The upwelled cold hypolimnetic water is shown near the coast of Erieau in satellite-based
observations and the 15 km resolution model (Fig. 8 a, b, e, f). The depth-averaged water temperature and current
circulation the in forecast results demonstrate that the up-welling process lasts several days (Fig. 11), with cold
hypolimnetic water accumulating along north shore and strong eastward currents along the northern shoreline of the
east central basin. The up-welling region matched that shown in a 2013 hindcast simulation (Valipour et al., 2019),
revealing the hotspots of vertical transport of nutrients and anoxic hypolimnetic water.
Another up-welling event occurred near the Cleveland drinking water intake crib on days 167-170 (Fig. 9).  This
event was accompanied by simultaneous ~8 mg L$^{-1}$ oscillations in the dissolved oxygen concentration (Fig. 9 c) at
STN 45164 (~20 km away from STN 45176), followed by the dissolved oxygen concentration remaining hypoxic (<
2 mg/L) for 2 days.  The COASTLINES model is shown to predict this event (section 3.2.2), which would provide
notice for drinking water plane operators to implement additional treatment required for hypoxic water.
Future work, using the embedded iWaterQuality module (formerly CAEDYM) could extend COASTLINES to
simulate biogeochemical parameters in Lake Erie (León et al., 2011), including dissolved oxygen (Bocianov et al.,

390 2020).

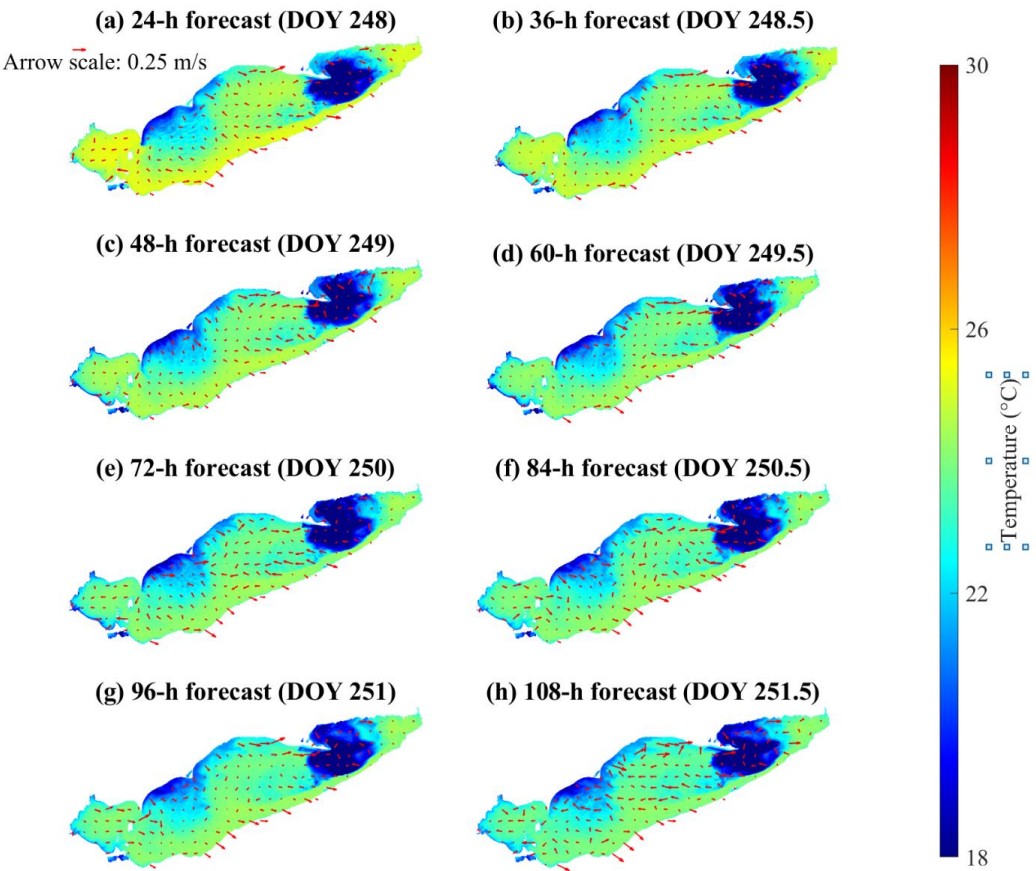

**Fig. 11 Color maps showing the forecast depth-averaged temperature throughout the lake. The red arrows represent forecast depth-averaged currents. The model results are from the 240-h forecast model initiated on day 247.**

### 4.3 Prediction of storm surge events for public safety

Due to its shallowness and long fetch aligned with the predominant southwest winds (Hamblin, 1979), Lake Erie has

the largest daily range of water level amongst the Great Lakes (Trebitz, 2006). In each month of 2020, Lake Erie set

new mean water level records (http://www.tides.gc.ca/C&A/bulletin-eng.html), causing the shoreline to be exposed

to high risk from erosion and flooding and making the shoreline communities susceptible to costly damage and

economic loss  (e.g. https://www.lowerthames-conservation.on.ca/flood-forecasting/flood-notices/). Given the

ability of COASTLINES to predict water level fluctuations induced by wind set-up (Fig. 3, 5), we test the ability of

the model to act as a storm-surge warning system.  This would assist early decision making during natural hazards

(Gronewold and Rood, 2019). Due to the unpredictability and severity of water level fluctuations in Lake Erie, there

is currently a need to improve short-term water level forecasts and water level warning systems (Gronewold and
Stow, 2014).
We forecast a storm event that occurred on 15 Nov. 2020, caused a dramatic water level increase (~1-1.5 m) in the
eastern basin with strong surface currents (Fig. 12). The inset image, taken during the event, shows flooding in
coastal areas. COASTLINES successfully predicted the phase of high-water level at Port Dover 72 hours in
advance, but underestimated the increase of water level with over 0.5 m. The forecast operated 24 hours accuracy in
water level prediction, with a difference <0.5 m from the observations (Fig. 12 d). Note that both forecasts missed
the small (~0.5 m) seiche before the significant increase at the end of day 320, presumably due to the low temporal
resolution of the meteorological forecast input or local topography near the gauge.
The hydrodynamic forecast output from COASTLINES could be further developed by enabling the coupled surface
wave model SWAN (Booij et al., 1999). Coupled Delft3D-SWAN models have recently been applied in the
development of a real-time predictive system for the coastal ocean and large estuaries (Rey and Mulligan, 2021).

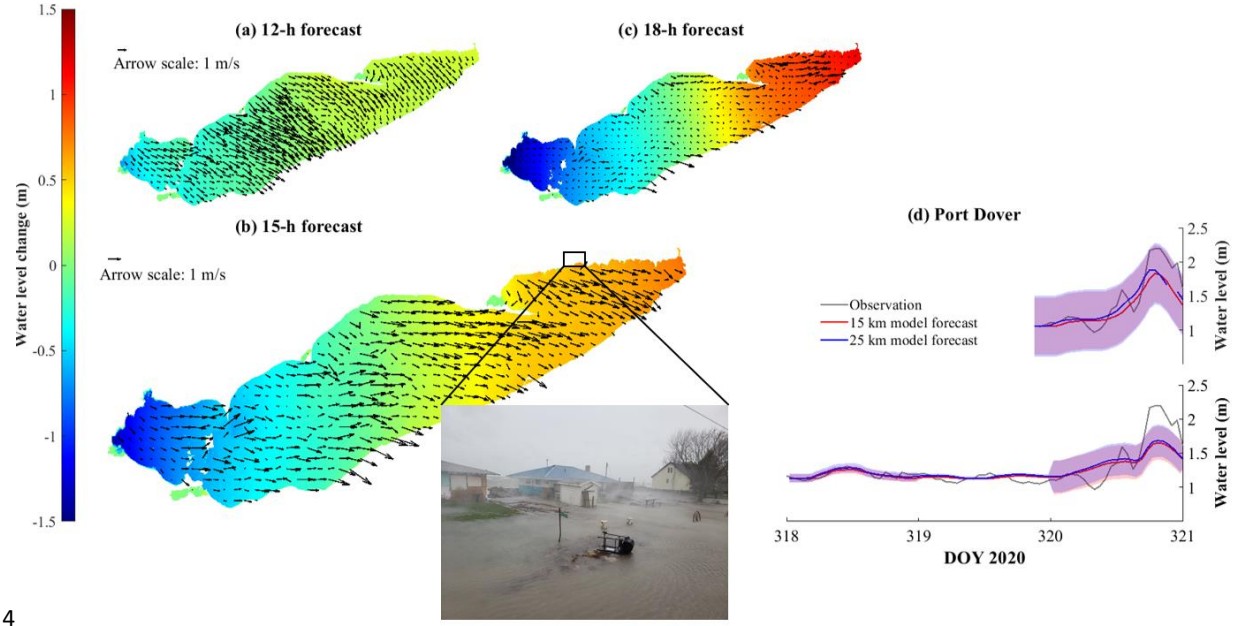

**Fig. 12 Color maps showing the water level change compared to Nov 15th 00h from (a) 12 h, (b) 15 h, and (c)**
**18 h forecasts from 15 km resolution model. The black arrows are depth-averaged mean current fields. Panel**
**(d) shows a comparison between forecast and observed water level at Port Dover. The upper panel shows the**
**24-h forecast, and the lower panel shows the forecast initiated on 12 Nov. 2020 (day 317). The shaded region**
**indicates the confidence interval. The inset image (extracted from a footage by J. Homewood from Lower**
**Thames Valley Conservation Authority) shows the flooding induced by the dramatic water level increase**
**during this event. The two cottages shown in the images were demolished later in the afternoon.**



## 5    Conclusions


We developed operational forecast system COASTLINES, using a Python-based wrapper code, to automate
application of the three-dimensional hydrodynamic model AEM3D to Lake Erie.  The resulting real-time and
predictive lake modelling system employs a processing chain that retrieves online meteorological forecast data,
prepares input files, executes the three-dimensional computational model and visualizes and compares model output
with observations on the web-based platform. This operational system shows the feasibility of applying freely
available meteorological forecasts, in situ buoy data and satellite images to drive and validate computational lake
models. The favorable agreement between forecast model results and observed physical variables (e.g., water levels
with RE ~ 40 % and temperatures with t-RMSD and s-RMSD < 2 °C) in Lake Erie demonstrates the ability of the
forecast system to make predictions of hydrodynamic processes on time horizons up to 240-h that are as accurate as
traditional hindcast simulations.
The near real-time updates to the web platform are an effective approach to rapidly disseminate forecast results to
stakeholders. Examples we have investigated include at least 24-h prediction of: (1) up- and down-welling events
that cause fish kills; up-welling events that bring hypoxic water to drinking water intake; and (3) coastal flooding
events from storm surges.
The global coverage of the GDPS weather model allows this system to be extended to other lakes and water
systems.  To facilitate further development of open-access predictive modelling systems, agencies are encouraged to
share observations in real-time through organizations such as GLEON (www.gleon.org) and GLOS (www.glos.us).
This will enable extension of COASTLINES to include prediction of the biogeochemical variables that drive
sediment transport, hypoxia and harmful algal blooms.

**Code and data availability.**

The observation data used in this study are openly accessible online, and cited and explained in the text. The forecast
model data can be obtained by contacting author Dr. Shuqi Lin (shuqi.lin@queensu.ca). The Python code used for
the COASTLINES were shown in the Appendices. The AEM3D can be installed and run with the licence purchased
from Hydronumerics (http://www.hydronumerics.com.au/), and its source code is available with permission from
Hydronumerics.

**Author contributions.**

The concept of the COASTLINES workflow was designed by LB, SL, SS, and RM, and SL carried them out. SL
developed the model code and performed the simulations. All authors contributed to the validation of the model and
interpretation of the results. SL wrote the manuscript with contributions from LB, SS, and RM.

**Acknowledgements.**

This project was funded by the Dean's Research Fund from the Faculty of Engineering and Applied Science at
Queen's University. Computational support .was provided by Alexander Rey and FEAS-ITS.  LB thanks Damien



Bouffard for discussions during visits to EAWAG, which inspired this research.  James Homewood, from the Lower
Thames Valley Conservation Authority (LTVCA) providing footages of the storm event on Nov. 15<sup>th</sup>, 2020.



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





## Appendix A: Code for retrieving observational data

### A1: Water level from gauges

```
from selenium import webdriver
import urllib
import os
import requests
import csv
from datetime import datetime
from bs4 import BeautifulSoup
import time
import matlab.engine
import calendar

dt = datetime.now()
last_day_of_month = calendar.monthrange(dt.year, dt.month)[1]
name_month = calendar.month_name[dt.month][:3]
Mon = str(dt.month)+'%2F'+str(last_day_of_month)

name_Bar_point = 'Bar Point_waterlevel_'+name_month+'.csv'
name_Kingsville = 'Kingsville_waterlevel_'+name_month+'.csv'
name_Erieau = 'Erieau_waterlevel_'+name_month+'.csv'
name_Colborne = 'Colborne_waterlevel_'+name_month+'.csv'
name_Dover = 'Dover_waterlevel_'+name_month+'.csv'
name_Stanley = 'Stanley_waterlevel_'+name_month+'.csv'

os.chdir('…\observation\water level')
try:
    os.remove(name_Bar_point)
    os.remove(name_Kingsville)
    os.remove(name_Erieau)
    os.remove(name_Colborne)
    os.remove(name_Dover)
    os.remove(name_Stanley)
except:
    os.chdir('…\observation\water level')

Barpoint_page
='https://marees.gc.ca/eng/Station/Month?type=1&sid=12005&tz=EST&pres=2&date=2020%2F'+ Mon
Kingsville_page =
'https://marees.gc.ca/eng/Station/Month?type=1&sid=12065&tz=EST&pres=2&date=2020%2F'+ Mon
Erieau_page =
'https://marees.gc.ca/eng/Station/Month?type=1&sid=12250&tz=EST&pres=2&date=2020%2F'+ Mon
Colborne_page =
'https://marees.gc.ca/eng/Station/Month?type=1&sid=12865&tz=EST&pres=2&date=2020%2F'+ Mon
Dover_page =
'https://marees.gc.ca/eng/Station/Month?type=1&sid=12710&tz=EST&pres=2&date=2020%2F'+ Mon
Stanley_page =
'https://marees.gc.ca/eng/Station/Month?type=1&sid=12400&tz=EST&pres=2&date=2020%2F'+ Mon

def retrieve_from_web(stationpage,stationname):
    page = requests.get(stationpage)
    page.raise_for_status()
    soup = BeautifulSoup(page.text)
    data = soup.find("div",class_="stationTextData")
```





```
with open(stationname, 'a') as csv_file:
writer = csv.writer(csv_file,lineterminator ='\n')
writer.writerow(['Date','Time','WL'])
for i in range(1,len(data.contents),2):
each_data = str(data.contents[i])
each_data = each_data.split()[1].split(";")
Date = each_data[0]
Time = each_data[1]
WL = each_data[2]
with open(stationname, 'a') as csv_file:
writer = csv.writer(csv_file,lineterminator ='\n')
writer.writerow([Date,Time,WL])
retrieve_from_web(Barpoint_page,name_Bar_point)
retrieve_from_web(Kingsville_page,name_Kingsville)
retrieve_from_web(Erieau_page,name_Erieau)
retrieve_from_web(Colborne_page,name_Colborne)
retrieve_from_web(Dover_page,name_Dover)
retrieve_from_web(Stanley_page,name_Stanley)
```



**A2: Lake buoy data acquisition examples**

```
from selenium import webdriver
import urllib
import os
import requests
import csv
from datetime import datetime
from bs4 import BeautifulSoup
import time

Month = datetime.now().month
Date = datetime.now().day

def data_NDBC(station_web,station_name):
    page  = requests.get(station_web)
    page.raise_for_status()
    soup = BeautifulSoup(page.text)
    each_line = soup.contents[0].split('\n')
    for i in range(0,2):
        each_data = each_line[i]
        with open(station_name , 'a') as csv_file:
            writer = csv.writer(csv_file,lineterminator = '\r')
            writer.writerow([each_data])
    for i in range(len(each_line)-2,2,-1):
        each_data = each_line[i]
        with open(station_name , 'a') as csv_file:
            writer = csv.writer(csv_file,lineterminator = '\n')
            writer.writerow([each_data])

## Station 45142 from NDBC
name_45142 = "STN 45142_" + str(Month) + "." + str(Date)+".csv"
os.chdir('…\\observation\\temperature\\STN45142')
try:
    os.remove(name_45142)
except:
    os.chdir('…\\observation\\temperature\\STN45142')
website1 = 'https://www.ndbc.noaa.gov/data/realtime2/45142.txt'
data_NDBC(website1,name_45142)

## Station 45167 from GLOS
def retrieve_45167():
    for name_45167 in glob.glob('.../Downloads/*45167*'):
        print name_45167
    try:
        os.remove(name_45167)
    except:
        os.chdir('…/14sl105/Downloads')
    driver = webdriver.Chrome()
    driver.get("https://glbuoys.glos.us/tools/export?data_type=buoy&units=eng&locs=45167")
    select_bottom = list()
    select_bottom.append("//*[@id='btn-clearParam']")
    select_bottom.append("//*[@id='Wind_Speed']")
    select_bottom.append("//*[@id='Wind_from_Direction']")
    select_bottom.append("//*[@id='Water_Temperature_at_Surface']")
    select_bottom.append("//*[@id='Air_Temperature']")
```





```
try:
for i in select_bottom:
elem = driver.find_elements_by_xpath(i)
elem[0].click()
except:
for i in range(0,len(select_bottom)):
elem = driver.find_elements_by_xpath(select_bottom[i])
elem[0].click()
download_bottom = "//*[@id='btn-download']"
elem2 = driver.find_elements_by_xpath(download_bottom)
elem2[0].click()
time.sleep(3)
driver.quit()
for name_45167 in glob.glob('…/Downloads/*45167*'):
print name_45167
f45167 = pd.read_excel(name_45167,header = 5)
Time = f45167[['Date/Time (UTC)']]
air_temp = (f45167[['Air_Temperature (fahrenheit)']]-32)*5/9
surf_temp = (f45167[['Water_Temperature_at_Surface (fahrenheit)']]-32)*5/9
wind_spd = (f45167[['Wind_Speed (kts)']])/1.944
wind_dir = f45167[['Wind_from_Direction (degrees_true)']]
select_column = pd.concat([Time,surf_temp,air_temp,wind_spd,wind_dir],axis= 1)
select_column.columns = ['Time in UTC','Surf_temp (C)','air temp(C)','wind speed (m/s)','wind direction']
select_column.to_csv(r'…\observation\temperature\STN45167\temp '+str(Month) + "." +
str(Date)+'.csv',index = None, header = True)
try:
retrieve_45167()
except:
print('Can not retrieve observations from buoy 45167')
```





**A3: Lake surface temperature from satellite imagery**

```
from selenium import webdriver
import urllib
import os
import requests
from datetime import datetime
from bs4 import BeautifulSoup
import time
YY = datetime.now().year
MM = datetime.now().month
DD = datetime.now().day
if DD<3:
MM = MM-1
if MM<10:
MM = "0" + str(MM)
else:
MM = str(MM)
page = "https://coastwatch.glerl.noaa.gov/erddap/files/GLSEA_GCS/" + str(YY) + "/" + MM + "/"
print(page)
driver = webdriver.Chrome()
driver.get(page)
def retrieve_satellite():
html = urllib.urlopen(page).read()
soup = BeautifulSoup(html, 'html.parser')
tags = soup('a')
nclst = list()
for tag in tags:
if tag.get('href').endswith('.nc'):
print(str(tag.get('href')))
nclst.append(str(tag.get('href')))
url = page + nclst[-1]
res = requests.get(url, allow_redirects=True)
print(res.raise_for_status())
os.chdir("…\observation\satellite data")
open(nclst[-1], 'wb').write(res.content)
time.sleep(3)
driver.quit()
try:
retrieve_satellite()
except:
print('No data today')
driver.quit()
```



**Appendix B: COASTLINE website snapshot**

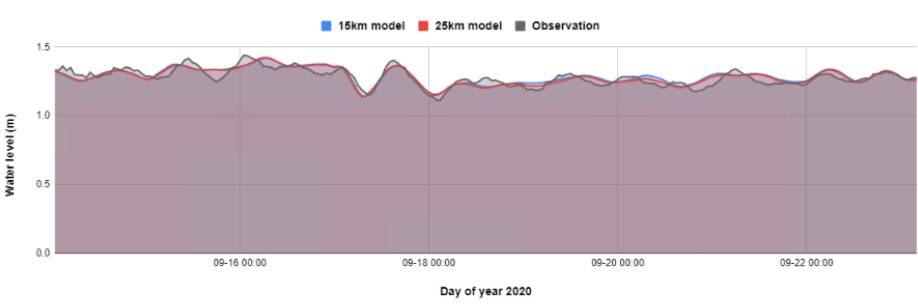

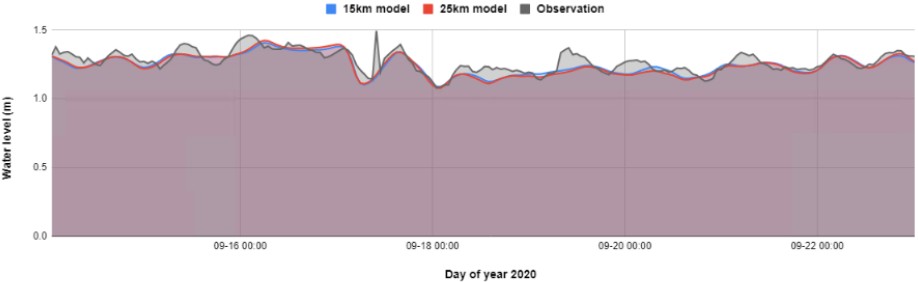


**Fig. B1 Snapshot of water level forecast validation web page displayed on COASTLINES online platform:**
**https://coastlines.engineering.queensu.ca/erie/water-level-forecast/. Status on Sep 23rd, 2020.**



**Appendix C: Validation of meteorological input variables**

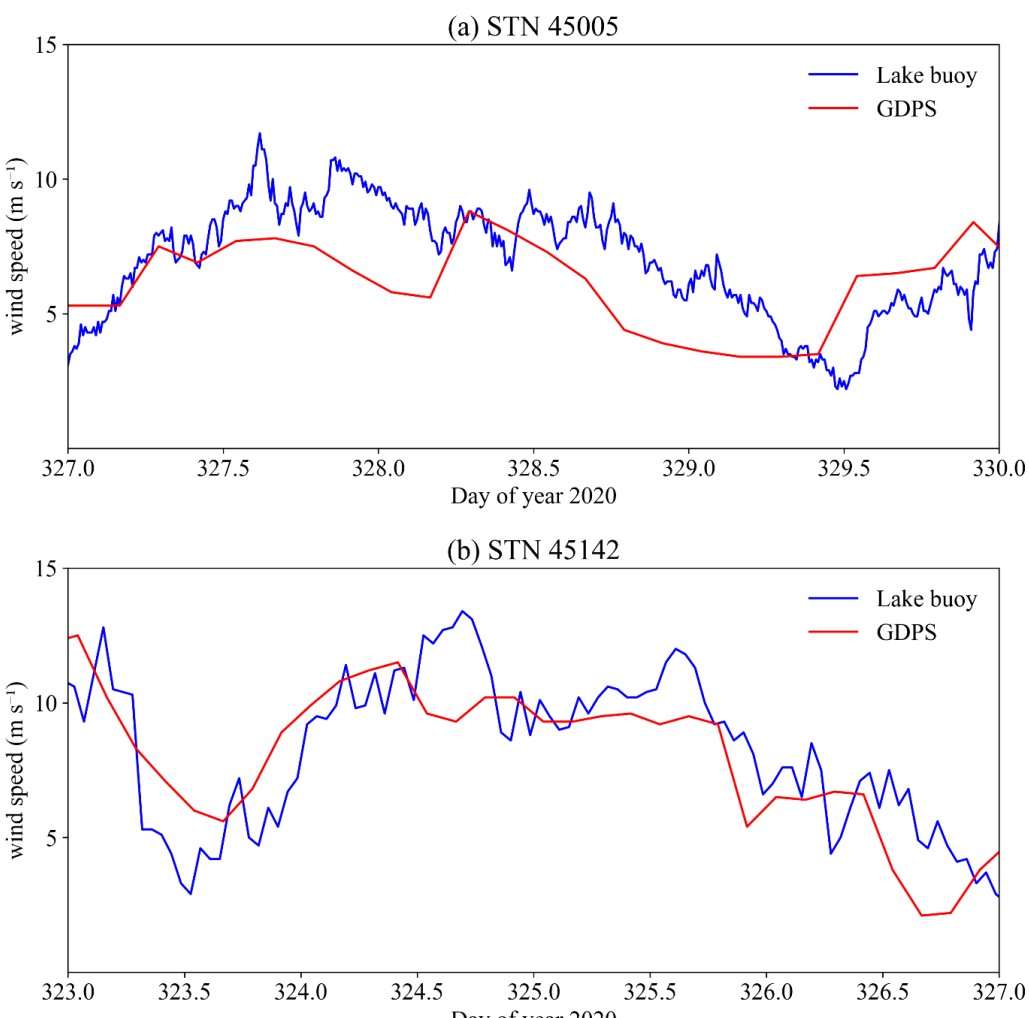


**Fig. C1 Comparisons of stitched GDPS wind forecast with 3 h delivery interval and lake buoy measured wind**
**speed at (a) station 45005 (10 min sampling interval), and (b) station 45142 (1 h sampling interval). The wind**
**gusts on day 327 at station 45005 and day 324 at station 45142 were missed by wind forecast.**



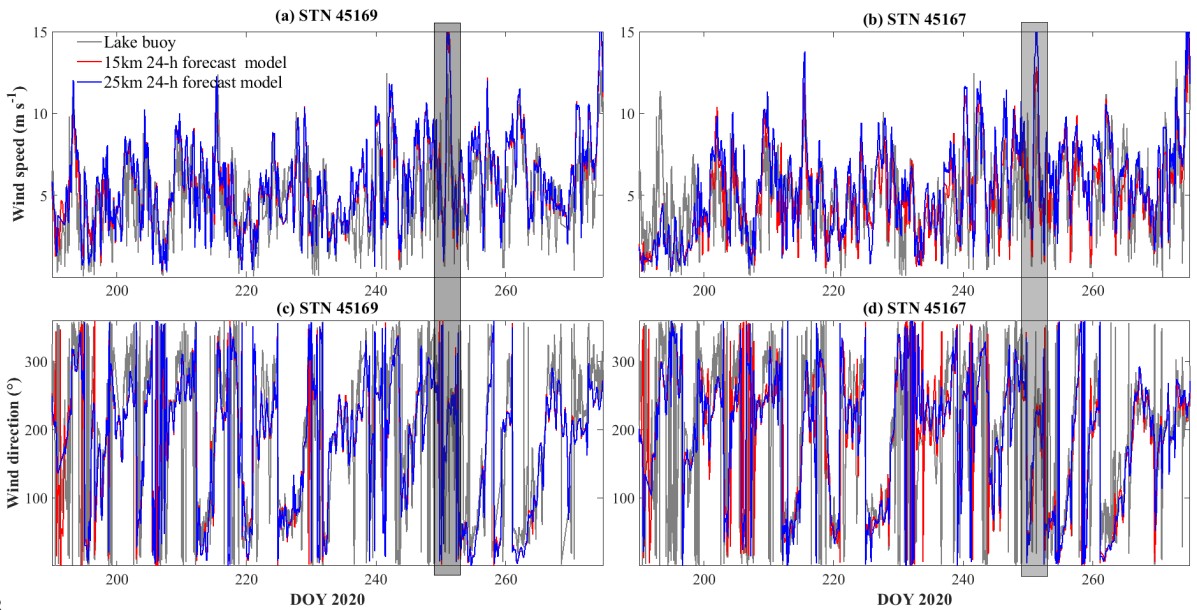

**Fig. C2 Comparisons of 24-h meteorological forecast and lake buoy observations of wind speed (a, b) and wind direction (c, d). The gray rectangle indicates the storm that led to up-welling along northern shoreline on days 248-253.**



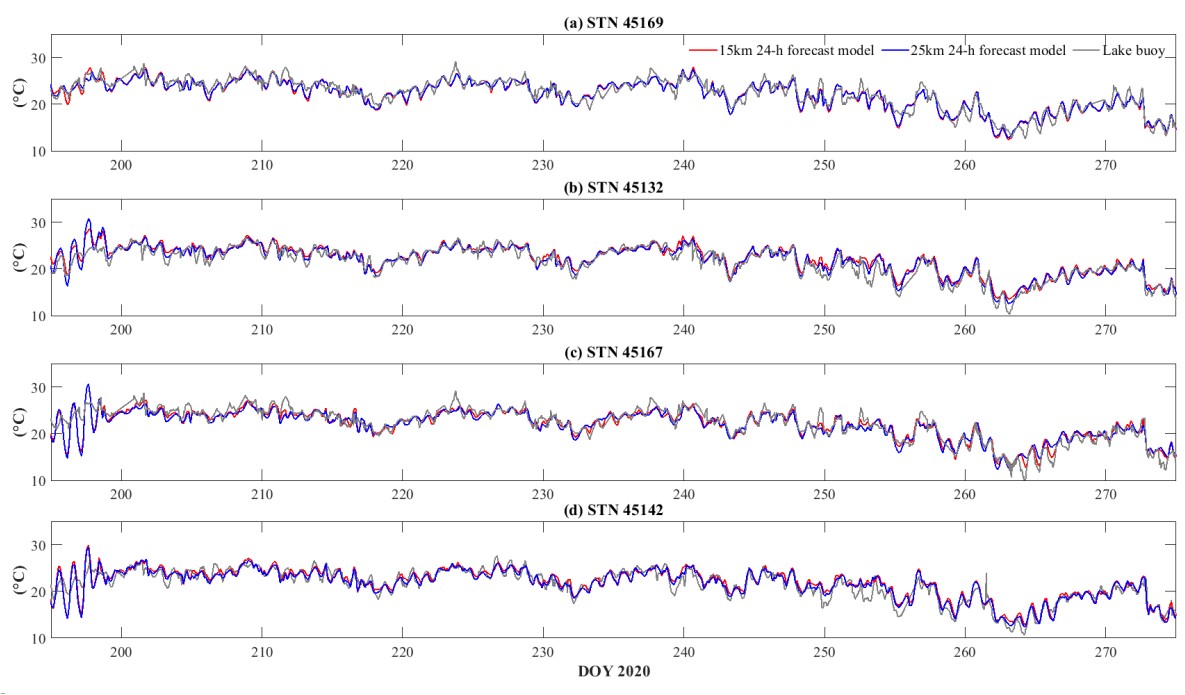

**Fig. C3 Comparisons of 24-h air temperature forecast and lake buoy observations of air temperature.**

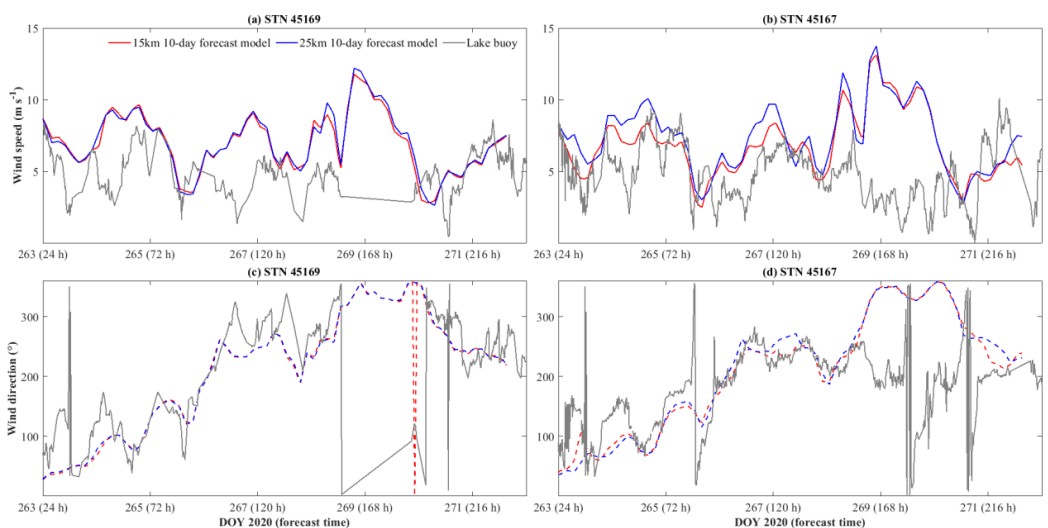


**Fig. C4 Comparisons of 240-h meteorological forecast and lake buoy observations of wind speed (a, b) and wind**
**direction (c, d).**



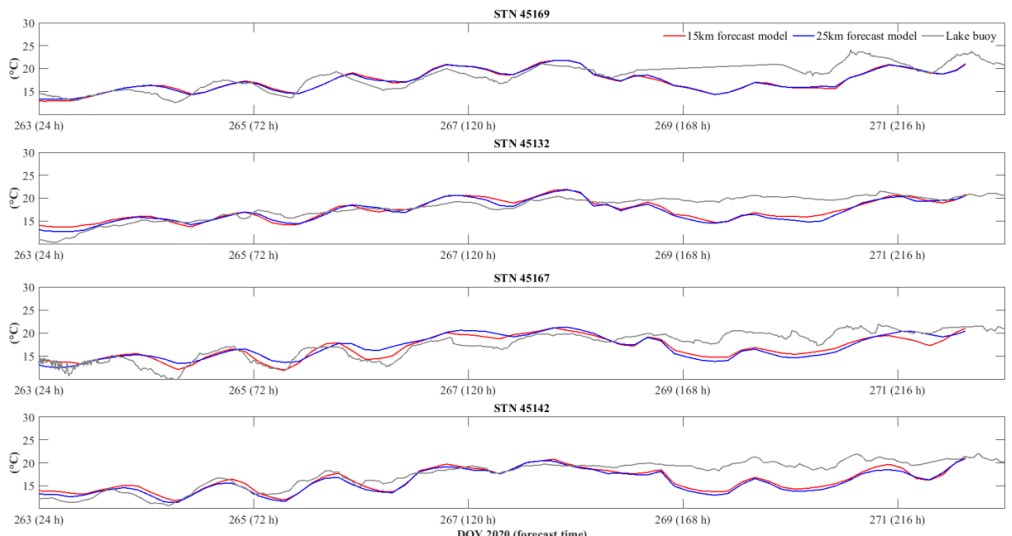


**Fig. C5 Comparisons of 240-h air temperature forecast and lake buoy observations.**



**Appendix D: Temperature validation against satellite observations**

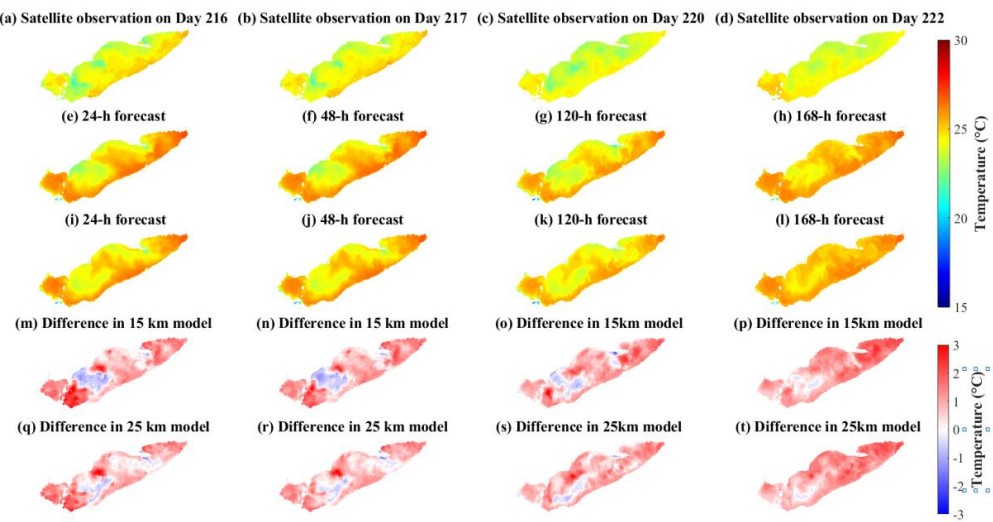


**Fig. D1 comparisons of (a-d) satellite observations, (e-h) 15 km model forecast, and (i-l) 25 km model forecast**
**during summer. The difference between observations and models are shown in (m-t).**

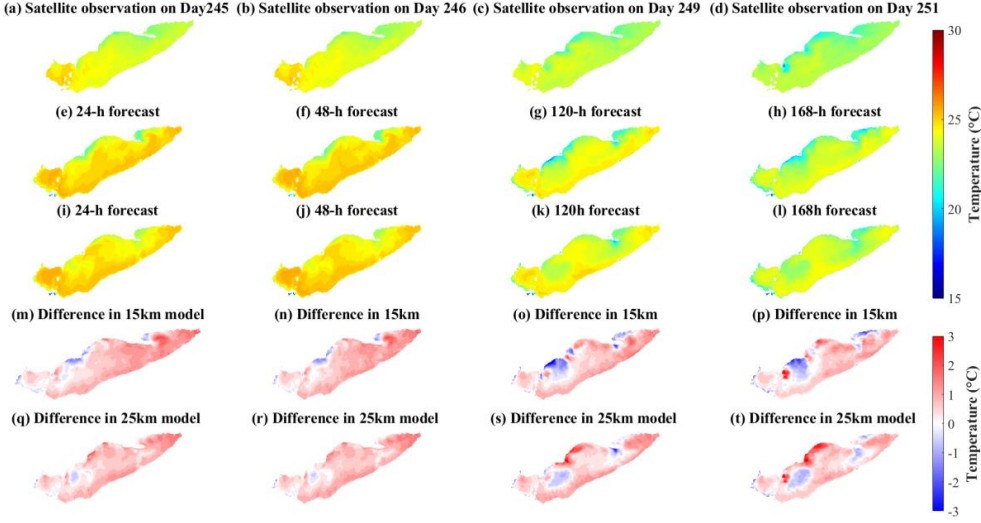


**Fig. D2 comparisons of (a-d) satellite observations, (e-h) 15 km model forecast, and (i-l) 25 km model forecast**
**during late summer. The difference between observations and models are shown in (m-t).**