# Peer review of "An automatic lake-model application using near real-time data"

_Geoscientific Model Development, 2021_

## Author Comment (AC1)

Dear Dr. Kerkweg,

Thank you for your suggestions and for notifying us about our mistake in the Code Availability Section.

As you suggested, we have added "COASTLINES" into the title of our article, and we have made the framework of COASTLINES open-source, by uploading all the executive code for running COASTLINES to the Dataverse (see Code and data availability). The code for running COASTLINES is now accessible at a doi (https://doi.org/10.5683/SP2/VTN7WC).

In the revision, we have adopted the suggestions from you and reviewers, stressing the novelty of the study, supplying the details in the COASTLINES workflow, and making the discussion about forecast result more consistent.

Thank you very much for your consideration.

Regards,

Shuqi Lin and co-authors

---

## Author Comment (AC2)

**RC2**

**General comments:**

This study develops an operational forecast system with a three-dimensional lake hydrodynamic model and validated the performance of hindcast experiments in Lake Erie. To conduct a robust forecast, the study separated 24-h and 240-h forecast simulations, and update the restart file for 240-h forecast every day. The data retrieval, numerical simulation, and validation are automated. This study is a development of a forecast system rather than a new model, but it is still of importance from the perspective of research implementation to society. However, some information for long-term steady operation is lacking, and providing us with such information would be helpful to other models/system developers, which are listed in the following major comments.

> **Reply 1:** Dear Dr. Tokuda, thank you for taking the time to review our manuscript, and your comments are very supportive of our work. In terms of long-term steady operation, if you are indicating the system stability in couples of months, we have added a comparison of stitched 24-h model forecast and continuous model run with 2 months of retrospective meteorological forecast in Appendix A (Fig. A1). So far, the system has been stably running under supervision over 2020-2021 without any runtime errors in the model and keps generating iterative forecast results every day. We have added the information about the supervisor of COASLTINES in lines 207-208,
>
> "The authors (supervisors of COASTLINES) and Queen's ITS monitor forecast results and maintain system operation."

**Specific comments:**

How did you set the initial value of the model? You mentioned that the model was 'cold started' on day 99 (line 116) and generates a restart file with a 24-h forecast simulation (line 185), but some other sentences imply that the model was initiated on another date (e.g., lines 288 and 299). I imagine that "initiated" means the model was just restarted in the consistent simulation (if so, only lead (forecast) time information is enough), but the authors are requested to clearly explain the system setup because such information is useful for other forecast systems without data assimilation.

**Reply 2:** Thank you for the correction and sorry for the confusion here. Your understanding is correct. To avoid the misunderstanding, we replaced 'initiated' with 'hot-started' in the revision (e.g., captions of Fig. 5, 8, and 10). We also modified the content in 2.5 System operation to clarify the approach (lines 188-191, lines 193-195):

"The model advances every day according to the 24-h forecast simulation and terminates by generating 're-start' files. These files are then used to hot-start the 240-h forecast simulation and the 24-h simulations for the next day. The input files for the 240-h forecast simulations are iteratively replaced by the new 240-h meteorological forecast generated each day."

"The long-term stability of employing daily 'hot' restarts can be seen in a comparison between simulated temperature profiles from a continuous run and that from stitching together the 24-h hot-start simulations (Appendix A; Fig. A1)."

What is the difference of the meteorological forcing data between 15km and 25km? If your system can work in time with 15-km data, I think you can focus on the results of 15-km data only because the difference of results seems to be negligible. Or is there any known problem with 15-km data?

**Reply 3:** 15 km and 25 km are the different horizontal resolutions of the meteorological forecasts. We wanted to test the sensitivity of the model to meteorological input resolution, but it turns out that the discrepancies in 15 km and 25 km meteorological forecast are minor, and for the large lake (e.g., Lake Erie), the forecast model is not sensitive to these minor discrepancies (Fig. 3, 7). The evaluation metrics (Table 2) show that the forecast outputs from 15 km model did not show lower deviation as expected, and sometimes even had higher bias. However, the 15 km model was more sensitive in predicting the upwelling event (Fig. 8, 9, D2). Thus, we suggest meteorological forecast with 15 km should be implemented to detect the meso-scale phenomena like upwelling. We have added the discussion in lines 452-454.

"The meteorological forecast from the 15 km and 25 km GDPS models did not show discrepancies (Fig. C2-5) and the evaluation metrics indicate that forecast results were largely insensitive to the meteorological inputs in Lake Erie (Fig. 3, 7). However, the 15 km model better predicted the mesoscale upwelling event (Fig. 8, 9, D2)."

You showed one-year results, but I have an interest in the long-term stability of the system operation. Do you reset the initial condition every year?

**Reply 4:** We did not reset the initial condition every year. So far, we have not seen any runtime error in the model for 2020-2021 winter and spring. Moreover, we have run continuous hindcasts with AEM3D over 2002-2012 and the model does not show significant drift.

2 Data and methods

2.2 Model description

Line 96: Which programming language is the AEM3D written in? You mentioned that the wrapper code is written in Python, but it has no advantage on computational efficiency. In addition, you pointed out that previous hydrodynamic models are difficult to apply to the hind- and forecast applications due to the computational cost in line 42, how did you solve the problem?

**Reply 5:** AEM3D is written in Fortran which has advantage on computational efficiency and the model itself can complete the short-term forecast (up to 10 days) at a reasonable real/run time ratio (0.5h for 24h simulation, and 4h for 240h simulation). However, we treat the hydrodynamic driver as a black box. We agree that Python is not computationally efficient, but we also pointed out that it is the complex and time-consuming setup and calibration procedure in the hindcast application of the model that limits the implementation of models for hydrodynamic forecasts (lines 42-47).

"In the case of hindcast applications, the complex and time-consuming setup and calibration procedure, of these models, can result in a significant time lag (months to years) between when a project is initiated and when the model results are communicated to stakeholders. This delay severely limits the utility of computational models for policy and management decision making. For better application of these powerful computational tools, the ability for rapid monitoring and simulation forecasts should be established."

The manuscript underlines the importance of the automatic workflow developed in this study in the Introduction, which could accelerate the data acquisition and model setup procedure, and eliminate the hurdle in generate iterative forecast results (lines 69-72).

"In the present study we developed and tested the COASTLINES (Canadian cOASTal and Lake forecastINg modEl System; https://coastlines.engineering.queensu.ca/) lake-model application workflow, that rapidly accesses near real-time online data (weather forecasts, water level and temperature observations) for automated model forcing, execution and validation. "

2.3 Model setup and meteorological forcing variables

Line133: I have three questions.

(1) The mass balance of lakes is described in + Precipitation – Evaporation + Riverine inflow – Riverine outflow ± Groundwater infiltration/seepage. Did you consider the groundwater component? If not, explicitly describe that assumption.

> **Reply 6:** No, we did not consider the groundwater component. Due to the large amount of water stored in Lake Erie and its large lake surface area, the hydrodynamic simulation of the lake usually does not include the groundwater component since it plays a minor part in the water circulation.

(2) In addition, is Precipitation – Evaporation balanced in Lake Erie? This budget controls the seasonal variability in water level, but mass imbalance may cause a problem in longterm operation.

> **Reply 7:** Yes, we agree that the water balance assumption may affect the model results in the long-term forecast. But in terms of the short time scale (10-day forecast) the oscillation of water level was mainly caused by surface seiches (Trebitz, 2006). The Precipitation – evaporation balance does not affect the water level significantly in such a short time scale. Unfortunately, there is no way to reset the water level within AEM3D using the restart file; therefore, in our projections we adjusted the predicted water level according to the real-time observed gauge levels. Future work will employ machine learning to forecast inflows/outflows so as to achieve a water balance.
> We added this information into lines 136-139,

"In this pilot application, the Lake Erie inflows and outflows, which roughly balance, are neglected, however evaporation and precipitation are accounted for in the water balance. Over short timescales (<10 days), the contributions from evaporation and precipitation to water level change are minor, with water level oscillations resulting from storm surges and surface seiches (Trebitz, 2006)."

(3) Even if the riverine in/outflows are balanced, can you ignore the effect on the fluid velocity field near the inlets and outlets? If you can, add the reference.

**Reply 8:** Yes. We ignored the effect on the fluid velocity filed near the river months. Because Lake Erie has such a large scale, this is a very small part of the model domain and we did not see this simplification to cause any errors or affect the hydrodynamic predictions, except the water level forecast near Bar point (lines ??).

3 Results

You showed the confidence shade in the figures of time series, how did you do conduct ensemble simulations? This question is related to the reliability of the system if it is operated as a warning system to society.

**Reply 9:** We did not conduct the ensemble simulations, but using the averaged statistical metrics gained from the previous forecast evaluation to indicate the confidence level. In the beginning of the Results, we have modified the text in lines 219-223 as

"The water level statistical metrics (RMSD and RE) were ensembled and averaged over April to September 2020. The 24-h and the 240-h forecast lake surface temperature and temperature profiles, from the models, were also validated against real-time lake buoy data and daily averaged satellite imagery. The timeseries and spatial MBD and RMSD (t-RMSD, t-MBD and s-RMSD, s-MBD) were ensembled and averaged over July to September 2020."

3.2.1 Lake surface temperature

Line 273: Those results are interesting; longer forecast time does not increase the error for surface temperature (thermodynamics) even with consistent bias according to Fig. 6 but does the error for water level (hydrodynamics) according to Fig. 4 and 5. Can you discuss the difference? Comparison between the model bias and forecast time would be helpful in this respect.

**Reply 10:** Thank you for pointing this out. The comparisons between model bias and forecast time have been shown in Fig. 3 and 7 for water level and lake surface temperature, respectively. The water level oscillations in Lake Erie are mostly due to the surface seiches (Trebitz, 2006), which depend largely on the wind, compared to the lake temperature, which depend mainly on the air temperature.   The wind forecast has larger uncertainty as the forecast time extends (Fig. C4, 5); thus, we can see the obvious growth of bias against forecast time in water level (Fig. 3) but not in lake surface temperature (Fig. 7). We have supplied some discussion on this topic (lines 450-452 and lines 460-461).

"The rapid response of the water level to windstorms (Hamblin, 1987) could result in the effects of aliasing and forecast error being passed to the water level, leading to the growth of RE against forecast time (Fig. 3)."

"The growing bias in air temperature, with forecast time,  does not affect the lake surface temperature (Fig. 7), presumably owing to the buffer effect of surface mixing layer (Schertzer et al., 1987). "

Line 280: Why could you conclude the underestimation is due to ignoring river inputs? The underestimation occurs on the east side of the inlet from the Detroit River. Can you have a consistent discussion between line 223 and here?

**Reply 11:** To make a consistent discussion, we have moved the discussion about bias induced by neglecting inflows into 4.3 (lines 462-469).

"Neglecting the inflows and outflows in the predictive simulation could induce bias in the forecast. The overestimation of water level fluctuation range near Bar Point (Fig. 4f) may result from neglecting the large Detroit River inflow, which regulates the seiche magnitude. The inflows also adjust more rapidly to air temperatures compared to deep lake waters. Thus, the up to 2 °C cold bias in coastal regions of the western basin (Fig. 8 m-t, Fig. D2) could be induced by neglecting the heated flux from two major inflows (i.e., Detroit River and Maumee River) of Lake Erie."

Fig. 8: What is the main reason for the consistent underestimation in lake surface temperature?

**Reply 12:** It is induced by the bias in air temperature forecast as the forecast lead time increase (e.g., Fig. C5). We have added the discussion in lines 456-458,

"The 168-h forecast meteorological data overestimated wind speeds by up to 10 m s$^{-1}$ (Fig. C4), and bias in the air temperature forecast (Fig. C5) may cause the consistent warm bias (up to 3℃) in forecast lake surface temperature (Fig. 8)."

4 Discussion

4.1 Bias and uncertainty

If the system developed in this study focuses on the forecast of some critical events like coastal up-welling and storm surge as discussed in the Sects. 4.1 and 4.2, could you move this Sect. 4.1 after the Sect. 4.3? The current Sect. 4.1 discussed the mean RMSD and compared it with a previous study incorporating data assimilation, but the data assimilation corrects only the initial condition of state variables in a model; not boundary conditions including meteorological forcing data. On the other hand, some of the critical events are caused by extreme atmospheric conditions in my understanding. So, can you discuss the model uncertainty and further improvement separating into initial-value and boundary-value problem?

> **Reply 13:** Thank you for the suggestion. We have moved Sect. 4.1 to the last part of Discussion (4.3) and separate the discussion into initial condition-induced errors and boundary condition-induced errors (lines 429-461).

Technical corrections:

2.3 Model setup and meteorological forcing variables

Line 116: How did you set the initial condition for the water level?

> **Reply 14:** AEM3D does not allow for adjustment of the water level in the restart file. Therefore, we project the changes in water level at the gauges, using the observed gauge value adjusted by the predicted water level change according to the model. This is done in post processing (within the automatic workflow). We have added the explanation in lines 195-198. Eventually the model will be further developed to compute a proper water balance.
>
> "At present, the initial water level cannot be modified using the AEM3D re-start files. Therefore, to account for long term drift in surface water level, we used real time gauge observations as the datum point for water level forecasts (automatically performed by MATLAB in post processing) and only consider

errors resulting from simulation of storm surges and seiches, as opposed to those from seasonal changes in mean lake level."

Line 119: What is "which is CFL = (Hodges et al., 2000)?

**Reply 15:** We are sorry about the typo here. It should be CFL = $\sqrt{2}$ (line 121).

Line 121: "and net longwave radiation" (the former one) -> "and downward longwave radiation"? Because net (downward -upward) longwave radiation is calculated within the model as you mentioned.

**Reply 16:** We have corrected the text here as you suggested.

3.2.1 Lake surface temperature
Line 252: Lake "S"urface temperature -> Lake "s"urface temperature

**Reply 17:** We have corrected the text here.

Line 288: needs a punctuation after "day 251".

**Reply 18:** We have added a punctuation here. Thank you for correction.

4.2 Prediction of coastal up-welling for fishery and drinking water management
Fig. 11:
(1) Correct the caption of the colorbar (remains selected?), and the same problem happened in Fig. D1.

**Reply 19:** We have corrected both figures.

[Figure]

**Fig. 11 Color maps showing the forecast depth-averaged temperature throughout the lake. The red arrows represent forecast depth-averaged currents. The model results are from the 240-h forecast model hot-started on day 247.**

(2) Can you show the observation data?

**Reply 20:** Because this figure presented as a depth-averaged temperature, there is no observation data to compare against.

4.3 Prediction of storm surge events for public safety

Fig. 12:

(1) Can you show the spatial distribution of the 24-h and 96-h forecast to compare with the time series in (d)?

**Reply 21:** As you suggested, we have added the map of water level change from 24-h forecast and 96-h forecast in the Appendix E.

[Figure]

**Fig. E1 Spatial distribution of water level change from forecasts hot-started on Nov 15th (a, b) and Nov 12th (c, d). The reference water level is the observation at Nov 15th 00:00. The black arrows are depth-averaged mean current fields. The black squares in the upper right corners of each map indicate the location of Port Dover (Fig. 12d).**

(2) According to (a) to (c), the forecasted water level is highly dependent on forecast time, and can you show the relationship between water level and forecast time? (It does not seem to be saturated even if the latest data is used according to (a)

  **Reply 22:** Yes. Due to the windstorms, the water level in Lake Erie is highly dependent on surface seiche, with period ~ 14 h (Trebitz, 2006;Mortimer, 1987). Panels (a) to (c) show the forecasted basin-scale surface seiche during a windstorm. The relationship between water level and forecast time was shown at one location (Port Dover), where we have footage showing the effect of flooding, in panel (d). Could you please specify what kind of information you want us to visualize?

**Reference**

Hamblin, P. F.: Meteorological forecing and water level fluctuations on Lake Erie, J. Great Lakes Res., 13, 436-453, 10.1016/S0380-1330(87)71665-7, 1987.

Mortimer, C. H.: Fifty Years of Physical Investigations and Related Limnological Studies on Lake Erie, 1928–1977, Journal of Great Lakes Research, 13, 407-435, https://doi.org/10.1016/S0380-1330(87)71664-5, 1987.

Schertzer, W. M., Saylor, J. H., Boyce, F. M., Robertson, D. G., and Rosa, F.: Seasonal Thermal Cycle of Lake Erie, Journal of Great Lakes Research, 13, 468-486, https://doi.org/10.1016/S0380-1330(87)71667-0, 1987.

Trebitz, A. S.: Characterizing seiche and tide-driven daily water level fluctuations affecting coastal ecosystems of the Great Lakes, J. Great Lakes Res., 32, 102-116, 10.3394/0380-1330(2006)32[102:CSATDW]2.0.CO;2, 2006.

---

## Author Comment (AC3)

**RC1**

Lin et describes and evaluates a forecasting system for predicting 3-D thermal structure of Lake Erie. The manuscript is an evaluation of a true forecasting system (i.e., it is evaluating a set of forecasts of the future rather than mimicking the forecasting process with historical data). It uses forecasted meteorology from Environment Canada Global Deterministic Forecast System to drive the model. The model used (AEM3D) had previously been applied to Lake Erie so the novelty of the paper is using the model with forecasted meteorology. I really appreciated the discussion sections on how the forecasting system could potentially be used to anticipate critical events for decision makers, emergency managers, and users of the lake. I visited the website for the project and it is indeed up-to-date. The paper evaluates the performance of the forecasts for a three-month period of time, although the manuscript discusses forecasts from outside this period. The authors highlight the automation provided by a Python script, but this seems to lack novelty (thought it appears to get the job done). The model is a standardly used model, downloading one meteorology source and converting to another is straightforward, and using a task manager to run a job is routine. The application to Lake Erie is new but the manuscript is introducing a named forecasting system (COASTLINES) with the only model development being a Python script used to execute the model and download observations (actually also Matlab scripts). Further, the code for running COASTLINES is not provided. Overall, the novelty of COASTLINES beyond the application to Lake Erie needs to be better motivated in the introduction.

> **Reply 1:** Thank you for the comments and advice. We have made the framework of COASTLINES open-source, by uploading all the code for running COASTLINES to the Dataverse (https://doi.org/10.5683/SP2/VTN7WC) (see Code and data availability). To apply a computational model to a lake requires laborious data sourcing and preparation, model setup and calibration. Thus, it is impossible to rapidly generate forecast output. The novelty of our study is to automate this lake-model application, using open data, enabling timely hydrodynamic forecasts that are communicated on a web platform. We agree that all the tools we have used are routine, but we have developed a workflow that combines them into a useful product and are the only operational forecast system that solely uses publicly available online data for model forcing. As we stated in the Introduction (lines 69-72),

"In the present study we developed and tested the COASTLINES (Canadian cOASTal and Lake forecastINg modEl System; https://coastlines.engineering.queensu.ca/) lake-model application workflow, that rapidly accesses near real-time online data (weather forecasts, water level and temperature observations) for automated model forcing, execution and validation. Hydrodynamic forecasts are automatically post-processed and posted on a web platform."

The framework of this operational forecast system can be widely applied in other water systems around the world owning to the spatial coverage of the meteorological data and proliferation of near real-time lake observation data. We have emphasized this point in the conclusion section (lines 490-493),

"This operational system shows the feasibility of applying freely available meteorological forecasts (e.g., GDPS, HRRR), in situ buoy data and satellite images to drive and validate any computational lake model (e.g., AEM3D, DELFT3D, GLM), without modifying the source code. The global coverage of the weather model allows generalization of model application to and lake or coastal domain."

The manuscript only uses observations to evaluate the model. It does not perform data assimilation as other forecasting system do. The workflow highlights the automation of the evaluation using the observations but does not offer a way that the evaluation is used to improve the model or the forecasts. Therefore, it isn't clear why the automated evaluation is necessary. It would be great to explore how a feedback between the evaluation and the forecasts can be developed. The forecasts lack a representation of uncertainty. Uncertainty in forecasts is increasingly the state-of-the-art. There is reference to uncertainty in the figures, but the manuscript does not describe how uncertainty is estimated. At minimum, the discussion needs to address the lack of uncertainty in the forecast and explore how it might be included in the forecasts.

**Reply 2:** We cannot use data assimilation for model forecasting, because we do not have observations in the future to assimilate. Rather, we could apply data assimilation to better calibrate the real-time model, from which we generate our forecasts. This has been done (e.g., (Baracchini et al., 2020a)) to reduce the RMSE temperature simulation of Lake Geneva from ~ 2 ℃ to ~ 1 ℃ by employing data assimilation that required ~ 1 month of computational time. As future work, we will improve the overall model calibration (which is not included in the COASTLINES forecast workflow) by employing

the same OpenDA calibration (https://www.openda.org/). This approach would be consistent with our philosophy to develop modelling tools that can be universally applied to hydrodynamic source codes.

We have discussed the limitations of implementing data assimilation in current COASTLINES (lines 429-441)

"The uncertainty and bias in the COASTLINES forecast results from error induced by the initial conditions at each hot-start, error in the meteorological forecasts and error in the numerical methods. These errors could be reduced by improving model calibration through data assimilation. For example, (Baracchini et al., 2020a) reduced the RMSE temperature simulation of Lake Geneva from ~ 2 °C to ~ 1 °C by employing a data assimilation routine; this would correspond to a <5% improvement in simulation of Lake Erie summer surface temperature. Before implementing data assimilation, the limitations of such a scheme must be considered: (*i*) The lack of observations in the future, makes data assimilation impossible for adjusting forecasts; (*ii*) data assimilation is computationally intensive (Baracchini et al., 2020a) required ~1 month of computational time, clearly not an option for operational forecasting); and (*iii*) data assimilation requires modification of the source code, which is not consistent with our philosophy to develop modelling tools that can be universally applied. Rather, future work will focus on adding real time model calibration (e.g., (Gaudard et al., 2019)), which is not presently included in the COASTLINES forecast workflow. For example, (Baracchini et al., 2020a) employed OpenDA (https://www.openda.org/) as a black-box wrapper to calibrate DELFT3D for Lake Geneva. This approach can be adapted to any other model."

We have tracked model error with forecast horizon. The present evaluation provides the root-mean-squared-deviation (RMSD), relative error (RE), and mean bias deviation (MBD) of the forecast results which indicate the quantitative uncertainty in the prediction according to the horizon (Fig. 3 and 7(a)). The automated evaluation reveals the probabilistic nature of the prediction results. The automated evaluations included in the COSTLINES workflow provide a quantitative confidence interval in terms of the predictive horizon, presenting how the uncertainty grows through time (Fig. 3, 7) and what maximum and minimum limits we should expect.

In the section 2.4 (lines 152-155), we describe how we calculated RMSD and RE for water level predictions:

"RMSD is the absolute error of the model against the observation. The difference between the observed daily minimum and maximum value was defined as the daily water level fluctuation range, where RE is the ratio between the RMSD and lognormal mean of daily range over April to September 2020. Given that our

study focusses on a 240-h forecast, RE is able to characterize the forecast bias, regardless of the instantaneous water level position."

And in lines 173-178, we describe how we calculate MBD for water temperature predictions,

"We quantified the temperature forecast capability using the statistical measures of RMSD (eq. 1) and Mean Bias Deviation (MBD):

$$MBD = 100 \ \frac{\frac{1}{N}\Sigma_{i=1}^{N}(x_i-y_i)}{\frac{1}{N}\Sigma_{i=1}^{N} y_i} \tag{1}$$

For the spatial MBD and RMSD (s-MBD and s-RMSD), $x_i$ and $y_i$ are the model and observed temperature in each grid, and $N$ is the total number of grids. For timeseries MBD and RMSD (t-MBD and t-RMSD), $x_i$ and $y_i$ are the model and observed temperature at each sample time, and $N$ is the total number of samples."

As we stated in lines 219-223, we use ensembled and averaged RMSD, RE, and MBD to represent the confidence interval.

"The water level statistical metrics (RMSD and RE) were ensembled and averaged over April to September 2020. The 24-h and the 240-h forecast lake surface temperature and temperature profiles, from the models, were also validated against real-time lake buoy data and daily averaged satellite imagery. The timeseries and spatial MBD and RMSD (t-RMSD, t-MBD and s-RMSD, s-MBD) were ensembled and averaged over July to September 2020."

And the estimation of uncertainty has been included in the application of forecast (e.g., Fig. 4, 5, and Fig. 12(d)). The shaded areas in the panel represent the confidence interval, indicating the highest water level we should expect in terms of the forecast time scale, according to the ensembled and averaged RMSE, RE, and MBD.

The argument for why data assimilation is not necessary could be stronger. They argue that it would only potentially decrease the RMSE by ~half (0.7C) by citing another study that used data assimilation. However, how does the reader know whether this is a meaningful magnitude?

**Reply 3:** We agree. See Reply 2 above. We have added further info on the magnitude of improvement (lines 429-441).

Instead of putting the Python script in the Appendix, I recommend putting them in a repository like Zenado. That would allow someone to use the scripts without having to cut and paste from

the Appendix. Furthermore, the manuscript highlights the Python code but also has a dependence on MatLab for foundational parts of the workflow (i.e., line 131-132 – the conversation of weather forecasts into the AEM3D input). Therefore, it is a Python + Matlab + Windows Schedule workflow. Additionally, the code that is provided is only for retrieving the observations. The code to run the forecasting system is missing. Overall, I do not think that the availability of code and model output meets GMD's standards.

> **Reply 4:** We agree and have posted the COASTLINES source code in the Dataverse (see Code and data availability). This includes the wrapping Python code, MATLAB scripts, and the timeline set in Windows task scheduler. The hydrodynamic driver is set as an executable black box and, while available form a third party, could be replaced with any executable hydrodynamic code.

The term hindcast is used throughout the manuscript but is not defined. It would be help to define exactly what a hindcast is.

> **Reply 5:** Thank you for the suggestion. We have added the definition in line 37-38:
> "Over the past several decades, many computer models have been applied to hindcast (running models forced with and validated against historically collected data) lake hydrodynamics to aid management."

Line 440 states "To facilitate further development of open-access predictive modelling systems", which is a major oversell of COASTLINES as being in the group of open-access predictive modeling. The hydrodynamic model requires a license and the code to run the forecasting system is not made available.

> **Reply 6:** The modelling system we developed has been placed in a data repository. The hydrodynamic driver (AEM3D) is treated as a black box (as often done with OpenDA for calibration, (Baracchini et al., 2020a)), is available from Hydronumerics and cold be replaced by any other executable hydrodynamic code.

Can the authors point to a manuscript that demonstrates that the re-start works or provide a figure that shows that a restarted run matches a run that was continuous (i.e., same length of simulation but without stopping and restarting)? Some hydrodynamic models are designed to be run from a cold-start and have internal variables that are not saved – thus preventing a true restart.

> **Reply 7:** To address your concern, we have added the temperature profile comparison of restarted run and continuous run in the Appendix A. It turns out that there is no obvious difference between these two modes (lines 207-209).

[Figure]

**Fig. A1 Temperature profile comparisons between (a) stitched 24 h model run with re-start files, and (b) model run with continuous input files.**

The manuscript highlights the use of the Environment Canada Global Deterministic Forecast System (GDPS) product. Could other freely available forecast model output be used? What about NOAA's Global Forecasting System or NOAA's Global Ensemble Forecasting system?

> **Reply 8:** Yes. In this study, the meteorological forecasts are provided by GDPS, but other weather forecast model with various horizontal resolutions can also be used (e.g., High Resolution Rapid Refresh (HRRR) from NOAA (lines 210-212), see also(Rey and

Mulligan, 2021)). We are presently developing COASTLINES for Lake Ontario with DELFT3D as the black box hydrodynamic code and NOAA forcing.  These options will be included as switches in the COASTLINES workflow.

**Specific comments**

Line 47: Are there 1-D water temperature forecasting systems that can also provide context in the introduction and discussion?

> **Reply 9:** As you suggested, we have added reference to a Simatrat application which is a near-realtime system (lines 55-56).
>
> "Similarly, meteolakes.ch (Baracchini et al., 2020b) applies Delft3D for short-term 3D forecasts (4.5 days) of four Swiss lakes and simstrat.eawag.ch (Gaudard et al., 2019) applies Simstrat for near-realtime 1D simulation of 54 Swisss lakes. "
>
> Some 1D models lake models (e.g., General Ocean Turbulence Model (GOTM, https://gotm.net), GLM (Hipsey et al., 2014), and Freshwater lake model (FLake, (Mironov, 2008)) have been incorporated with climate forecast system (e.g., Climate Forecast System (CFS) at US National Centers for Environmental Prediction (NCEP; Saha et al., 2014), European Centre for Medium-Range Weather Forecast (ECMWF; Johnson et al., 2019)) to conduct retrospective forecast of water temperature in the lakes (Mercado-Bettín et al., 2021;Lv et al., 2019), but none of them are operational forecasting systems. Also, these implementations require relatively complex processes of input data (climate forecast outputs) acquisition and re-analysis, hindering their usage in real-time. We also provided some comparison of the RMSD of LST in an 1D model (Freshwater Lake [FLake]) and in COASTLINES (lines 425-428)
>
> "COASTLINES also outperforms 1D climatological hindcasts (e.g., Freshwater Lake (FLake)), with 2– 4 °C RMSD over a 120-h lake surface temperature forecast (Lv et al., 2019;Gu et al., 2015) and has similar error to the 3D Princeton Ocean Model (Kelley et al., 1998), with 0.6-0.9 °C mean absolute error in the 36-h lake surface temperature forecast at station 45005."

Line 63: I recommend starting a new paragraph here.

      **Reply 10:** We started a new paragraph here as you suggested.

Line 95: Is there a specific version of the AEM3D that was used? Without that the forecasting system is not reproducible.

      **Reply 11:** We are using AEM3D version 1.1.1. The information was supplied in line 108.

Line 119: what is CFL =?

      **Reply 12:** We are sorry about the typo here. CFL = $\sqrt{2}$ (line 121)

Line 156: What happens when there are run-time errors?

      **Reply 13:** We used the try/except function in Python to avoid the program accidently stopping due to run-time errors (See the code archived in the Dataverse). Also, the supervisor of the COASTLINES system routinely monitors the whole system for troubleshooting (lines 207-208).

      **"**The authors (supervisors of COASTLINES) and Queen's ITS monitor forecast results and maintain system operation.**"**

Line 184: Is the restart file used to generate a 216-hr forecast since the first 24-hr have already been generated?

      **Reply 14:** Yes. Your understanding is correct. Because 24h and 240h forecast are driven by same 10-day meteorological data, there is no need for repeating the first 24h forecast.

Line 189: The Windows Task Scheduler is another dependency of the forecasting system. Can the forecasting system only run in a Windows environment?

**Reply 15:** It is straightforward to run the forecasting system operationally in Unix-like operating systems (e.g., Linux Mint, Ubuntu, macOS) by using the software utility *cron*, which is a time-based job scheduling tool that will run jobs/scripts at regular intervals.

Line 224: The sentence refers to the estimation of uncertainty in the model forecast but the manuscript lacks methods describing the uncertainty estimation process.

**Reply 16:** Sorry for the confusion here. We quantify the forecast uncertainty in terms of the Root Mean Square Deviation (RMSD), Relative Error (RE), and Mean Bias Deviation (MBD), and the calculation processes were mentioned in section 2. 4 (Eq. 1-3). The uncertainties we showed in the Fig. 3-5 are statistical metrics of water level RMSD and RE ensembled over April to September 2020, and the uncertainties we showed in the Fig. 7 are the statistical metrics of LST MBD and RMSD ensembled over July to September 2020 (lines 219-223).

Line 260: What forecast horizon do these numbers refer to?

**Reply 17:** These numbers refer to 24h forecast as we indicated in the Table 2. We also added this information here (lines 274-275).

Line 329: Readily what? (missing word)

**Reply 18:** We are sorry about the typo here. The sentence should be
"the objective of the present work is to develop a simple automated lake modelling system that can be readily operated to diverse field sites to suit management needs."
This sentence has bee removed in the revision.

Fig 2: What is Phenomena detection (Supervisor)? It is not mentioned anywhere in the manuscript.

**Reply 19:** The authors and Queen's ITS play the supervisor roles, maintaining system normal operation and detecting the phenomena of interest. We have added the

explanation in lines 207-208: "The authors (supervisors of COASTLINES) and Queen's ITS monitor forecast results and maintain system operation."

Fig 2: The caption says "Daily Python workflow" but the text states that Matlab is also used.

**Reply 20:** We modified the caption and deleted "Python" here.

**Reference**

Baracchini, T., Hummel, S., Verlaan, M., Cimatoribus, A., Wüest, A., and Bouffard, D.: An automated calibration framework and open source tools for 3D lake hydrodynamic models, Environmental Modelling & Software, 134, 104787, 2020a.

Baracchini, T., Wüest, A., and Bouffard, D.: Meteolakes: An operational online three-dimensional forecasting platform for lake hydrodynamics, Water Res., 172, 10.1016/j.watres.2020.115529, 2020b.

Gaudard, A., Vinnå, L. R., Bärenbold, F., Schmid, M., and Bouffard, D.: Toward an open-access of high-frequency lake modelling and statistics data for scientists and practitioners. The case of Swiss Lakes using Simstrat v2.1, Geosci. Model Dev., 12, 3955-3974, 10.5194/gmd-2018-336, 2019.

Gu, H., Jin, J., Wu, Y., Ek, M. B., and Subin, Z. M.: Calibration and validation of lake surface temperature simulations with the coupled WRF-lake model, Climatic Change, 129, 471-483, 2015.

Kelley, J. G. W., Hobgood, J. S., Bedford, K. W., and Schwab, D. J.: Generation of Three-Dimensional Lake Model Forecasts for Lake Erie, Weather and Forecasting, 13, 659-687, 10.1175/1520-0434(1998)013<0659:GOTDLM>2.0.CO;2, 1998.

Lv, Z., Zhang, S., Jin, J., Wu, Y., and Ek, M. B.: Coupling of a physically based lake model into the climate forecast system to improve winter climate forecasts for the Great Lakes region, Climate Dynamics, 53, 6503-6517, 10.1007/s00382-019-04939-2, 2019.

Mercado-Bettín, D., Clayer, F., Shikhani, M., Moore, T. N., Frías, M. D., Jackson-Blake, L., Sample, J., Iturbide, M., Herrera, S., French, A. S., Norling, M. D., Rinke, K., and Marcé, R.: Forecasting water temperature in lakes and reservoirs using seasonal climate prediction, Water Research, 201, 117286, https://doi.org/10.1016/j.watres.2021.117286, 2021.

Mironov, D. V.: Parameterization of lakes in numerical weather prediction: Description of a lake model, DWD, 2008.

Rey, A., and Mulligan, R. P.: Influence of Hurricane Wind Field Variability on RealTime Forecast Simulations of the Coastal Environment, J. Geophys. Res. Oceans, 126, 10.1029/2020JC016489, 2021.

---

## Author Response (AR2)

**Comments from Editor**

Dear Shuqi,

I have had some additional reviewer comments for you to consider. Overall they are positive about the article and I believe the code is of practical use, however some confusing sections on **data assimilation and uncertainty** were identified that should be addressed prior to publication. In my view, it would be fine to **outline current limitations of the system and future potential** as suggested by the reviewer without impacting the rational to publish.

Cheers,

Jeff

Dear Jeff,

Thank you for the time and comments on our manuscript. In the revision, we have made some clarification about data assimilation and uncertainty (Please see the responses to reviewers and changes in the manuscript).

Best,

Shuqi and co-authors

**Comments from Reviewer**

I thank the authors for their revisions. The key revisions include the archiving of the code and the addition of a hot-start analysis. While most of my concerns have been addressed, I do have some feedback on the revisions and the revision letter

> **Reply:** Thank you for your time and comments. We made some modifications and clarifications according to your advice. Please see them bellow and in the revision.

1) The revision with track changed did not have the changes tracked so it was difficult to compare to the original document. Can they upload a version with track changes?

> **Reply:** We uploaded a version with track changes.

2) In the revision letter (Reply 6), the authors state "The hydrodynamic driver (AEM3D) is treated as a black box (as often done with OpenDA for calibration, (Baracchini et al., 2020a)), is available from Hydronumerics and could be replaced by any other executable hydrodynamic code". The idea that the executable could be replaced by "any" another executable suggests that the framework is very general. Other executables likely differ in the format of the input driver data. Does the code provided convert the format of weather forecast to the input format of "any other" model? What other steps would be required to get the forecasting system to work with "any other" executable?

> **Reply:** The code we provided here can automatically download the daily forecast and convert the necessary input variables into a .dat file for AEM3D. However, since the input format in different models varies, the user must convert these input files to the

required formats for their model if they are not going to use AEM3D. For example, we are using this approach with DELFT3D and AED-GLM.

We added some explanation in lines 216-218.

3) The authors seem to be slightly confused about "data assimilation". In the revision letter they state: "We cannot use data assimilation for model forecasting, because we do not have observations in the future to assimilate." This statement is repeated on Line 436 in the revisions and needs to be removed. You don't need data in the future for data assimilation – instead you take yesterday's forecast of today and today's observations to run an assimilation using today's data. This then sets the initial conditions (and parameters if they are included in the data assimilation) for a forecast that starts today. This forecast, observation, assimilate, and forecast cycle is done routinely in meteorological forecasting and does not require future observations. Also, since sequential assimilation only requires assimilating the new data, it is not as computationally intensive as the authors claim. Yes, re-calibrating a model using a long-time series of would be computationally intensive but that is not what is required for sequential data assimilation methods like the ensemble Kalman filter. Operational weather forecasting uses data assimilation for much larger models than here, so it is entirely feasible. Finally, why does data assimilation require modifying the source code? In the sentences following, the authors state that routines like OpenDA could be used without modifying the code. Overall, this section about data assimilation is muddled due confusion about what data assimilation means and how it can be used. I am not asking the authors to do data assimilation but they should not use an incorrect argument to justify why to not use data assimilation. It would be more useful to lay out a road

map for what would be required to use a sequential DA method (like the Ensemble Kalman

Filter) in their modeling framework.

> **Reply:** Thank you for clarifying our confusion about data assimilation. We were
>
> referring to the procedure of nudging hindcasts to match observations (e.g., NARR). We
>
> removed the statement about the computational requirement for data assimilation. In
>
> AEM3D, users are not able to easily modify restart files, which are in binary format,
>
> without access to and modification to the restart file read/write statements in the source
>
> code (aem3d_restart_v3_type.f90). That is why we cannot easily adjust the initial
>
> conditions via data assimilation. As we mentioned in lines 442-452, the road map could
>
> be using the tool like OpenDA, which has been applied to DELFT3D, to implement data
>
> assimilation.

4) The "The water level statistical metrics (RMSD and RE) were ensembled and averaged" is

confusing because the term ensemble in forecasting is often meant to represent a set of forecasts

of the same period that differ in initial conditions, parameters, weather inputs, etc. Perhaps

remove the word "ensemble."

> **Reply:** Thank you for correcting. We removed "ensemble" to avoid confusion.

5) The authors state "And the estimation of uncertainty has been included in the application of

forecast". This is confusing because uncertainty is not included in a particular forecast. Instead,

the uncertainty is in the distribution when different forecast data are combined. The manuscript

needs to clarify that uncertainty is in the evaluation statistic from combining forecast dates – not

actual uncertainty in an individual forecast.

**Reply:** Thank you for the suggestion. We clarify the term "uncertainty" in lines 155-157.

---

## Author Response (AR3)

Dear Shuqi,

Thank you for your response to the reviewers and revised manuscript. I'm broadly happy except for one set of revisions. Like the reviewer I didn't understand the section on data assimilations and I still don't understand the revised version. It might be a terminology issue but I think this could be clearer.

"Before implementing data assimilation in our system, the limitations of such a scheme must be considered: (i) As we are performing forecasts, not hindcasts, we are unable to assimilate observations during model runtime (e.g., as done in the NCEP North American Regional Reanalysis: NARR); The lack of observations in the future, makes data assimilation impossible for adjusting forecasts;"

I think the confusion here is that you are mixing data assimilation for re-analysis with data assimilation for forecasting. In the latter you assimilate the observations in near real-time from which the forecast is made. This this "The lack of observations in the future, makes data assimilation impossible for adjusting forecasts" is simply not true as written.

"(ii) data assimilation is computationally intensive, required ~1 month of computational time (Baracchini et al., 2020a), clearly not an option for operational forecasting);"

I'm not sure why a sequential method would be necessarily more expensive. Again I think this relates to the above confusion.

"and (iii) Sequential assimilation could be employed to nudge the initial conditions for the 24-h runs with real-time-observed data."

Nudge one way of doing it but formally you are probably updating the model states

"This could be achieved by modifying the binary AEM3D restart files using model specific read/write statements in our Python workflow (e.g., from aem3d_restart_v3_type.f90), followed by smoothing (e.g., with a Kalman filter). "

I don't get the logic here, why would you update the restart file and then apply a smoother? Surly the Kalman filter (or more likely ensemble filter of some type for a nonlinear model) generates the updated states that are then written to the restart file. Also if AME3D needs multiple states initialised this could be very complicated in practice, which I think you mention in the response but not in the text.

That's a rather long way of saying I think you need to revise this statement a little more for accuracy. In my view you could implement a data assimilation scheme, subject to all the many challenges associated with that, and that you haven't is simply because its beyond the scope of the article and what you set out to do.

However, subject to this refinement I'm delighted to accept the manuscript given that this is not core to the research you present.

Best wishes,
Jeff

Dear Jeff,

Thank you for the explanation here and sorry for the confusion about the data assimilation. Since this part is beyond the scope of our paper, we decided to delete the statement about the limitations of data assimilation in the COASTLINES scheme, and instead state that

> "In AEM3D, sequential data assimilation could be implemented through modification of the restart files (aem3d_restart_v3_type.f90); however this is beyond the scope of the present study."

Again, thank you for providing the suggestions and accepting our manuscript.

Best wishes and Merry Christmas!

Shuqi and co-authors